# Emergence of SARS-CoV-2 subgenomic RNAs that enhance viral fitness and immune evasion

Harriet V. Mears[1], George R. Young[1,2,¤a], Theo Sanderson[3], Ruth Harvey[4], Jamie Barrett-Rodger[1,5], Rebecca Penn[1], Vanessa Cowton[6], Wilhelm Furnon[6], Giuditta De Lorenzo[6,¤b], Margaret Crawford[7], Daniel M. Snell[7], Ashley S. Fowler[7], Anob M. Chakrabarti[1,8], Saira Hussain[1,¤c], Ciarán Gilbride[1], Edward Emmott[9], Katja Finsterbusch[10], Jakub Luptak[11], Thomas P. Peacock[12,¤d], Jérôme Nicod[7], Arvind H. Patel[6,13], Massimo Palmarini[6,13], Emma Wall[14,15], Bryan Williams[15], Sonia Gandhi[16], Charles Swanton[5], David L. V. Bauer[1,13]*

1 RNA Virus Replication Laboratory, The Francis Crick Institute, London, United Kingdom, 2 Bioinformatics and Biostatistics STP, The Francis Crick Institute, London, United Kingdom, 3 Malaria Biochemistry Laboratory, The Francis Crick Institute, London, United Kingdom, 4 Worldwide Influenza Centre, The Francis Crick Institute, London, United Kingdom, 5 Cancer Evolution and Genome Instability Laboratory, The Francis Crick Institute, London, United Kingdom, 6 MRC-University of Glasgow Centre for Virus Research, Glasgow, United Kingdom, 7 Genomics STP, The Francis Crick Institute, London, United Kingdom, 8 UCL Respiratory, Division of Medicine, UCL, London, United Kingdom, 9 Centre for Proteome Research, Department of Biochemistry, Cell and Systems Biology, Institute of Systems Molecular and Integrative Biology, University of Liverpool, Liverpool, United Kingdom, 10 Immunoregulation Laboratory, The Francis Crick Institute, London, United Kingdom, 11 MRC Laboratory of Molecular Biology, Cambridge, United Kingdom, 12 Department of Infectious Disease, St Mary's Hospital, Imperial College London, London, United Kingdom, 13 Genotype-to-Phenotype (G2P-UK) National Virology Consortium, London, United Kingdom, 14 Crick/UCLH Legacy Study, The Francis Crick Institute, London, United Kingdom, 15 University College London and National Institute for Health Research (NIHR) University College London Hospitals (UCLH) Biomedical Research Centre, London, United Kingdom, 16 Neurodegeneration Biology Laboratory, The Francis Crick Institute, London, United Kingdom

¤a Current address: MRC London Institute of Medical Sciences, London, UK
¤b Current address: Area Science Park, Trieste, Italy
¤c Current address: WHO Collaborating Centre for Reference and Research on Influenza, Melbourne, Australia
¤d Current address: The Pirbright Institute, Surrey, UK
* david.bauer@crick.ac.uk

## Abstract

Coronaviruses express their structural and accessory genes via a set of subgenomic RNAs, whose synthesis is directed by transcription regulatory sequences (TRSs) in the 5′ genomic leader and upstream of each body open reading frame. In SARS-CoV-2, the TRS has the consensus AAACGAAC; upon searching for emergence of this motif in the global SARS-CoV-2 sequences, we find that it evolves frequently, especially in the 3′ end of the genome. We show well-supported examples upstream of the Spike gene—within the nsp16 coding region of ORF1b—which is expressed during human infection, and upstream of the canonical Envelope gene TRS, both of which have evolved convergently in multiple lineages. The most frequent neo-TRS is within the coding region of the Nucleocapsid gene, and is present in virtually all viruses from the B.1.1 lineage, including the variants of concern Alpha, Gamma, Omicron and descendants thereof. Here, we

**Data availability statement:** All relevant data are within the paper and its Supporting information files, or within the following repositories: Gel and blot images are available on Figshare (https://doi.org/10.25418/crick.27953013), numerical data are available on Figshare (https://doi.org/10.25418/crick.27952842), nanopore sequencing data analysis code is available on Figshare (https://doi.org/10.25418/crick.27959910), and amplicon sequencing data analysis code is available on Zenodo (https://zenodo.org/records/14277568). All nanopore and amplicon sequencing data files are available from EBI ArrayExpress (accession numbers E-MTAB-14681 and E-MTAB-14680).

**Funding:** This work was supported by the Francis Crick Institute which receives its core funding from Cancer Research UK (CC2166, CC1283), the UK Medical Research Council (CC2166, CC1283), and the Wellcome Trust (CC2166, CC1283). This work was also supported by the UK Medical Research Council (MR/W005611/1, MR/Y004205/1 to DLVB) and by UK Research and Innovation (to JL) and the Wellcome Trust (210918/Z/18/Z to TS). The Legacy Study is supported by the NIHR University College London Hospitals Biomedical Research Centre. The funders had no role in the study design, data collection and analysis, decision to publish, or preparation of the manuscript.

**Competing interests:** I have read the journal's policy and the authors of this manuscript have the following competing interests: While the authors declare no competing interests directly related to this work, CS receives grants from Bristol Myers Squibb, Ono Pharmaceuticals, Boehringer Ingelheim, Roche-Ventana, Pfizer, and Archer Dx; receives personal fees from Genentech, the Sarah Canon Research Institute, Medicxi, Bicycle Therapeutics, GRAIL, Amgen, AstraZeneca, Bristol Myers Squibb, Illumina, GlaxoSmithKline, MSD, and Roche-Ventana; holds stock options in Apogen Biotech, Epic Biosciences, GRAIL, and Achilles Therapeutics; is a member of a scientific advisory board for Bicycle Therapeutics, GRAIL, Relay Therapeutics, SAGA Diagnostics, and Achilles Therapeutics; is a co-founder of Achilles Therapeutics; receives consulting fees from Genentech, Medicxi, MetaboMed, Novartis, the China Innovation Centre of Roche, and the Sarah Cannon Research Institute; and receives honoraria from Amgen, AstraZeneca, Bristol Myers Squibb, Illumina, and Incyte. DLVB

demonstrate that this TRS leads to the expression of a novel subgenomic mRNA encoding a truncated C-terminal portion of Nucleocapsid, which is an antagonist of type I interferon production and contributes to viral fitness during infection. We observe distinct phenotypes when the Nucleocapsid coding sequence is mutated compared to when the TRS alone is ablated. Our findings demonstrate that SARS-CoV-2 is undergoing evolutionary changes at the functional RNA level in addition to the amino acid level.

## Introduction

SARS-CoV-2 has continued to evolve since its emergence in the human population [1]. An important emphasis throughout the pandemic has been on characterising the amino acid substitutions in new variants, particularly within the Spike glycoprotein, which contribute to increased transmission and immune evasion [2–4]. However, there has also been substantial evolution at the nucleotide level in both coding and non-coding regions of the genome [5].

Coronaviruses have polycistronic positive-sense RNA genomes, which contain numerous *cis*-acting RNA elements that regulate the viral lifecycle (Fig 1A). For example, translation of the first open reading frame, ORF1ab, is regulated by an RNA element which stimulates ribosomal frameshifting [6]. The subsequent ORFs, encoding the viral structural and accessory proteins, are translated from subgenomic messenger RNAs (sgmRNAs), which are synthesised via a mechanism known as discontinuous transcription, a form of programmed RNA recombination (Fig 1B) [7]. This is directed by corresponding transcription regulatory sequences (TRSs) located in the genomic 5′ UTR (leader, TRS-L) and upstream of each subsequent ORF (body, TRS-B) (Fig 1C) [8–11]. During negative strand synthesis, the viral RNA-dependent RNA polymerase complex may pause at a TRS-B, allowing the nascent RNA strand to dissociate and reanneal with the complementary TRS-L, then reinitiate transcription. The shared leader sequence in the 3′ of the negative-strand subgenomic RNA then serves as a promoter for positive-strand sgmRNA synthesis, producing a nested set of 5′ and 3′ co-terminal sgmRNAs (Fig 1Biii).

There are numerous factors which influence coronavirus sgmRNA expression levels. First, since discontinuous transcription takes place during negative strand synthesis, the most 3′ TRS-Bs are encountered more frequently than those more 5′, creating a gradient in sgmRNA expression, with the most 3′ ORF (encoding Nucleocapsid, N) being the most highly expressed. Coronaviruses may also alter sgmRNA expression levels by increasing or decreasing the degree of homology between the regions flanking the TRS-L and the TRS-B—or, more precisely, the strength of base-pairing between the TRS-L and the anti-TRS-B in the nascent negative strand RNA [12–14] (Fig 1C). This, along with long-range RNA–RNA interactions which may promote proximity to the TRS-L [12,15], enables highly regulated expression of coronavirus structural and accessory genes.

Analyses of sequencing datasets have identified new TRS-B sequences in SARS-CoV-2, which may lead to the expression of novel sgmRNAs [16–18]. In particular, the nucleotide substitutions G28881A, G28882A and G28883C, which underly the N:R203K,G204R mutation, create a consensus TRS (AGGGGAAC → AAACGAAC). These mutations define the B.1.1 lineage and its descendants, which includes three Variants of Concern: Alpha (B.1.1.7), Gamma (P.1) and Omicron (B.1.1.529 and its descendants). The presence of this new TRS-B was initially reported by Leary and colleagues [16] and the resultant sgmRNA was subsequently identified in publicly available RNA sequencing datasets [17] and RNAseq from infected cells [18]. However, deep sequencing of coronavirus-infected cells [19–21], and nidovirus-infected cells more broadly [22–26], often reveals a multitude of non-canonical

receives grants, paid to their institution, from AstraZeneca and GSK related to COVID-19, and is a member of the UK Genotype-to-Phenotype 2 Consortium. All other authors have declared that no competing interests exist.

**Abbreviations:** AUC, area under the curve; DMEM, Dulbecco's Modified Eagle Medium; IFN, interferon; SFB, short fragment buffer; TRSs, transcription regulatory sequences.

subgenomic RNA species. While some of these transcripts may be functional, and may be very highly expressed even from non-canonical TRS-like sequences [27], many are simply defective genomes from erroneous RNA recombination and can be overemphasised due to sequencing biases [28]; differentiating these possibilities from sequencing data alone is challenging. It is therefore still unclear to what extent this novel sgmRNA is expressed, whether it has a functional role during infection and if wider sgmRNA emergence is a general feature of SARS-CoV-2 evolution in the human population.

We therefore examined TRS-B emergence in SARS-CoV-2 in more detail. Here, we analyse global SARS-CoV-2 sequences to determine the prevalence of TRS-B evolution. We then focus on the novel TRS-B within the N gene, quantifying its expression in both cell culture and human swab samples. We further show that the protein product encoded by this new sgmRNA is expressed during infection, acts as an innate immune antagonist and contributes to viral fitness in cell culture. We find that mutation of the TRS-B within N attenuates viral replication in a distinct manner to mutation of the N coding sequence.

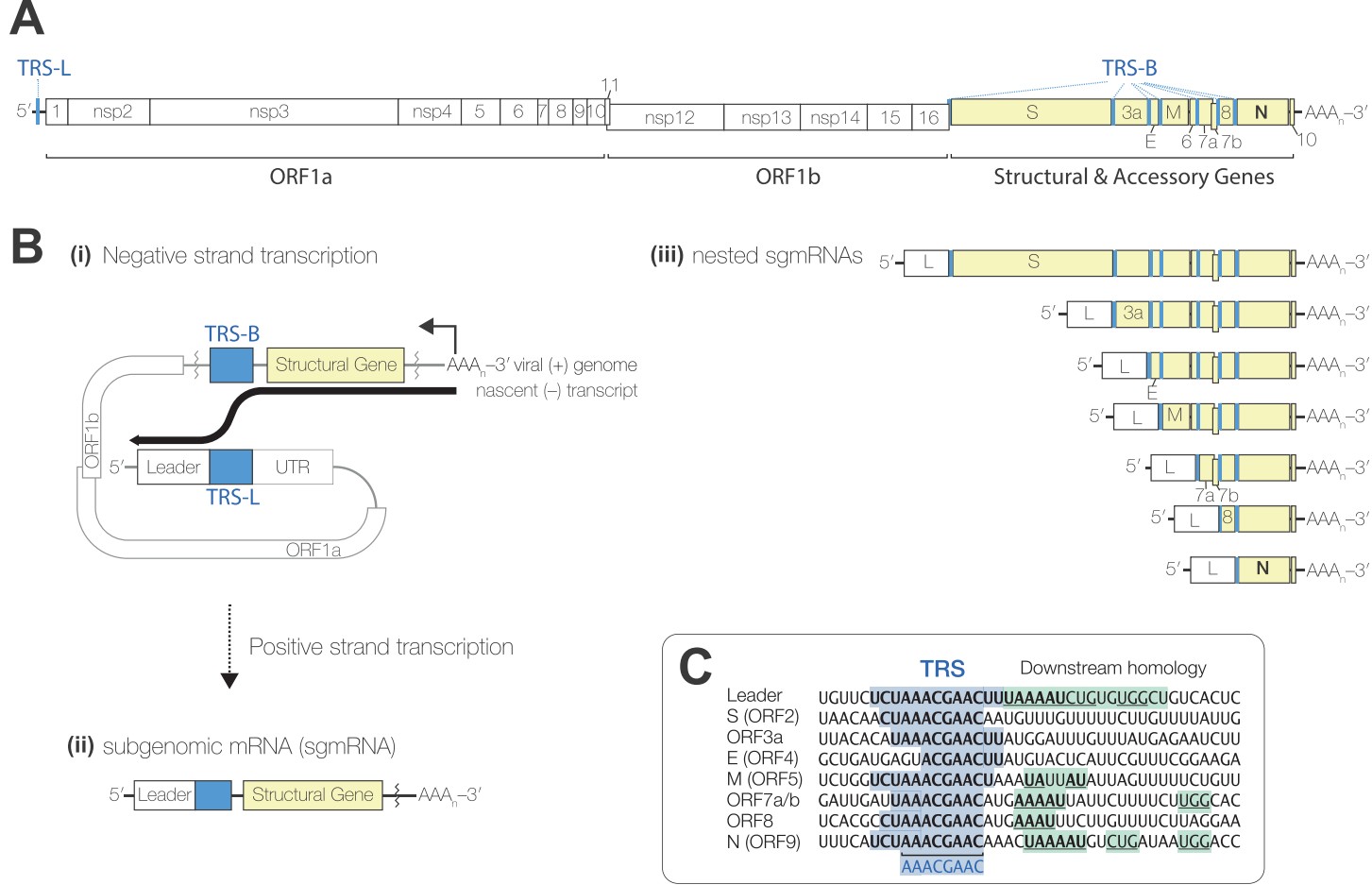

**Fig 1. Discontinuous transcription in coronaviruses. (A)** Schematic of the SARS-CoV-2 genome and **(B)** the mechanism of discontinuous transcription for expression of the structural (S, E, M, N) and accessory genes (yellow highlight). **(C)** Alignment of TRS-Bs upstream of each of the indicated ORFs, compared to the TRS-L. Regions of homology to the TRS-L are highlighted in blue and downstream homology to the 5′UTR is highlighted in green. TRS, transcription regulatory sequence.

## Results

### Detection of novel TRS-B sites

We set out to determine the frequency with which new TRS-B sequences have emerged in the global SARS-CoV-2 population. We searched the GISAID sequence repository for acquisition of the consensus TRS motif for SARS-CoV-2, AAACGAAC, relative to the Wuhan-Hu-1 reference sequence. Newly emerged TRS-Bs were non-uniformly distributed across the genome, and clustered towards the 3′ end (Fig 2A).

The most frequent of these neo-TRS-Bs is the G28881A, G28882A, G28883C mutant (N:R203K,G204R), described above (Fig 2A, red box). To date, this mutation is present in over 60% of SARS-CoV-2 sequences and, since the emergence of the Omicron variant and its sub-variants, is nearly fixed in the global SARS-CoV-2 population (>98% of sequences in the past 6 months, Fig 2B) [29,30]. This TRS-B is flanked by substantial sequence homology to the 5′ Leader (4 nt directly adjacent to the TRS, plus distal structures, Fig 2C and 2D). The resultant sgmRNA contains a start codon in frame with the N open reading frame at Met-210, which is in good Kozak sequence context for translation initiation (Fig 2D) [31]. This creates a new open reading frame that encodes residues 210-419 of N, fully encompassing the C-terminal and N3 domains (Fig 2C). Since this is the third internal open reading frame identified within the N coding region of SARS-CoV-2, we have named it 'N internal ORF 3′ (N.iORF3), following the naming system used by Finkel and colleagues [32]; N.iORF1 and N.iORF2 are alternative names for ORF9b, an internal out-of-frame ORF which encodes an innate immune antagonist.

To verify whether the N.iORF3 sgmRNA is transcribed during infection in cell culture, we infected Vero E6 cells at a high multiplicity with either a very early 2020 UK isolate (B lineage, before the evolution of the S:D614G mutation), a late 2020 UK isolate (Alpha variant, B.1.1.7 lineage), a late 2020 South African isolate (Beta variant, B.1.351 lineage), or a late 2021 UK isolate (Omicron BA.1 variant) and harvested cells at 24 h post-infection. RNA was extracted and analysed by reverse transcription PCR (RT-PCR) using a forward primer against the 5′ Leader and a reverse primer against the 3′-end of N (Fig 2E). When analysed by agarose gel electrophoresis, products corresponding to the canonical full-length N sgmRNA were detected in all SARS-CoV-2 infected cells, and an additional shorter product corresponding to the N.iORF3 sgmRNA was detected in Alpha- and Omicron-infected cells, but not lineage B-, and Beta-infected cells (Fig 2F). The identity of these PCR products was confirmed by nanopore sequencing (S1 Fig).

We then sought to confirm the presence of the N.iORF3 sgmRNA in human infections. We analysed occupational health screening swab samples from UK National Health Service (NHS) healthcare workers and employees of the Francis Crick Institute processed by the Crick COVID-19 Consortium Testing Centre and retained for analysis as part of the Legacy study (Cohort A1, NCT04750356). The Legacy study was approved by London Camden and Kings Cross Health Research Authority Research and Ethics committee (IRAS number 286469) and is sponsored by University College London Hospitals. We analysed 12 samples from each of the main COVID-19 "waves" from late 2020 to early 2022: B.1.177 (EU1, autumn 2020); B.1.1.7 (Alpha, winter 2020/1); B.1.617.2 (Delta, summer 2021); and BA.1 (Omicron, winter 2021/2). Analysis by RT-PCR revealed that N.iORF3 sgmRNA was specifically present in human clinical samples from the B.1.1 lineage (Alpha, Omicron) but not from non-B.1.1 lineages (EU1, Delta) (Fig 2F).

We next examined whether the N.iORF3 sgmRNA might have evolved independently outside of the B.1.1 lineage. We used Taxonium (see Materials and methods) to search public SARS-CoV-2 genomes within the complete UShER phylogenetic tree [33] that contained the

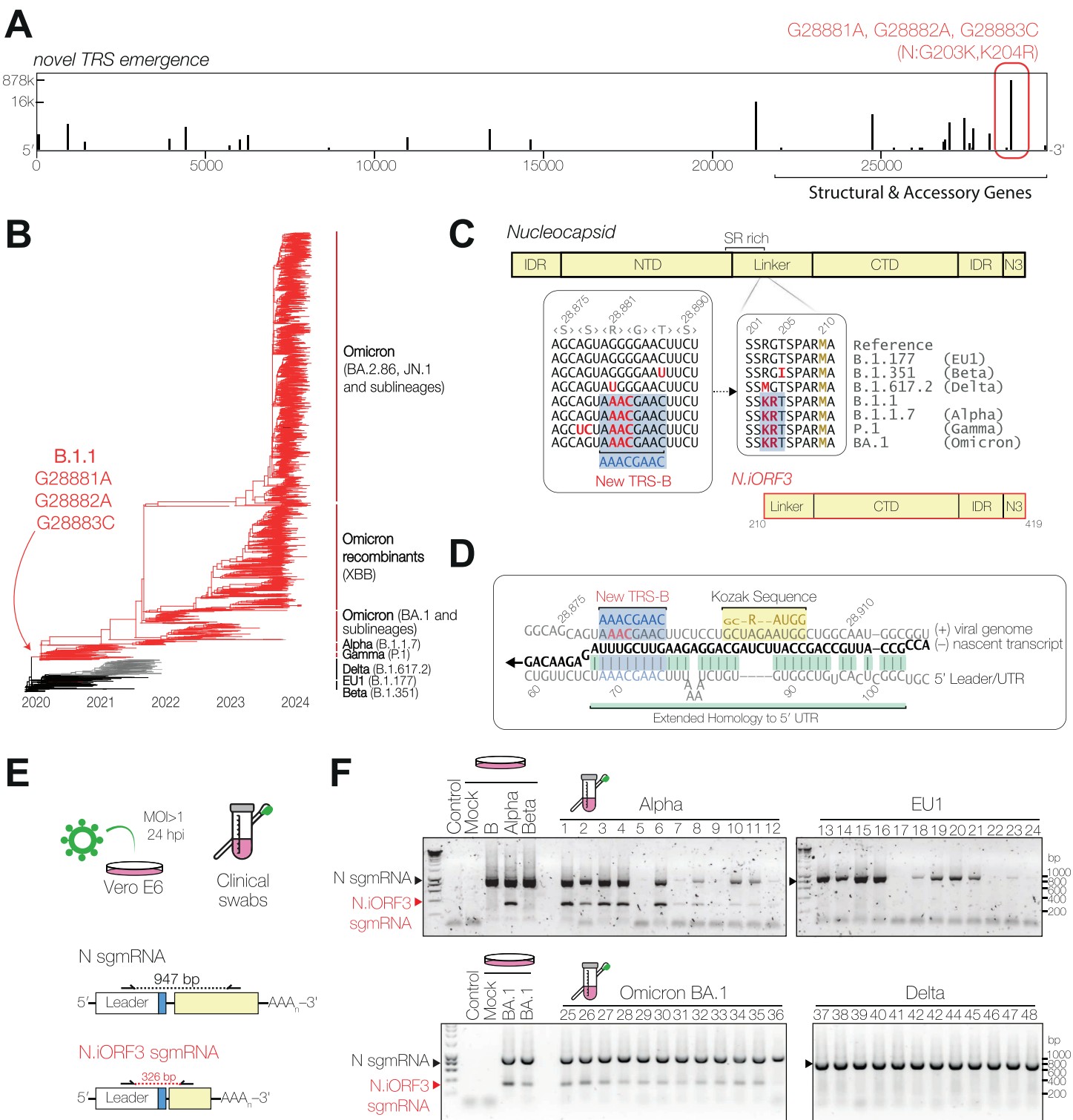

**Fig 2. Nucleocapsid R203K, G204R mutations in the SARS-CoV-2 B.1.1 lineage generate a novel TRS-B site and new subgenomic mRNA (sgmRNA). (A)** Frequency of emergence of the TRS-B consensus sequence (AAACGAAC) in the global SARS-CoV-2 population. The most frequent neo-TRS-B, within the nucleocapsid coding region, is highlighted in red. **(B)** Phylogenetic reconstruction of SARS-CoV-2 evolution in humans, with lineage-defining mutations for B.1.1 indicated. Viruses with N:R203K mutation (B.1.1 and its descendants) are coloured in red. Adapted from Nextstrain [103,104]. **(C)** Diagrams of nucleocapsid (N, top) and N.iORF3 (bottom) protein domains and sequence alignment of nucleotides 28874–28891, and amino acids 200–215 of N, showing emergence of a new TRS-B motif. Mutations relative to reference are highlighted in red. The N-terminal (NTD), linker, C-terminal (CTD) and N3 domains of nucleocapsid are shown, along with intrinsically

disordered regions (IDRs), and the serine-arginine (SR) rich region. **(D)** The sequence context of the novel N.iORF3 TRS-B (blue highlight), with extended base-pairing to the 5′UTR (green highlight) during (–) strand RNA synthesis (black), and downstream start codon with Kozak context (yellow highlight). **(E)** Schematic representation of reverse transcription PCR (RT-PCR) analysis of RNA extracted from infected VeroE6 cells at 24 h post-infection or nasopharyngeal swabs, showing positions of primers. **(F)** RT-PCR detection of canonical nucleocapsid (N) and N.iORF3 sgmRNAs in RNA extracted from infected cells and clinical swabs (numbered 1–48), for SARS-CoV-2 variants as indicated. Data underlying this figure can be found in: https://doi.org/10.25418/crick.27953013.

N:R203K,G204R mutation, but lay outside of the B.1.1 lineage. We found examples of convergent evolution of the N.iORF3 sgmRNA, notably within the Iota variant (B.1.526, S2A Fig); Iota was a Variant of Interest which circulated predominantly in the United States between late 2020 and summer 2021. These samples clustered within geographic regions, showed evidence of ongoing transmission, and were detected by multiple depositing laboratories, supporting their authenticity (S1 Table). Intriguingly, we also observed multiple independent instances of further evolution of the N.iORF3 TRS region (A28877U, G28878C) that increases homology to the 5′UTR (S2B Fig), notably in the entire Gamma lineage, at least six times within the Alpha lineage (1.3% of all sequences), and within Omicron, particularly in the BA.1 sublineage (2.1% of all sequences) ( S2C and S2D Fig and S1 Table) [29,30]. Sequences which have evolved the A28877U, G28878C mutations almost exclusively also carry the N.iORF3 TRS-B mutations, (>99.9%, S2E Fig), indicating that the upstream homology evolves after the N.iORF3 TRS-B emerges.

After N.iORF3, the most frequent novel TRS-B is located at the end of ORF1ab, within the nsp16 coding region, 251 nt upstream of the Spike ORF (Fig 3A). A two nucleotide substitution at positions C21304A and G21305A (underlying nsp16:R216N) creates a consensus TRS-B immediately upstream of four start codons (Fig 3B and 3C). The first, second and fourth start codons are in the +1 reading frame, in moderate Kozak sequence context, and would encode a short transframe peptide that we have designated nsp16.iORF1; the third start codon, also in moderate Kozak context, is in frame with the main nsp16 ORF and encodes a C-terminal portion of nsp16, designated nsp16.iORF2.

We then searched archived SARS-CoV-2-positive clinical swabs from the Legacy study that had been sequenced, and found six with the nsp16 R216N mutation, collected between September 2020 and January 2021. We analysed RNA extracted from these swabs by RT-PCR using primers in the 5′ leader and in the 5′ portion of Spike coding region (Fig 3D). We observed amplification of the canonical Spike sgmRNA in all samples, and also observed a longer nsp16.iORF-specific product in four samples, while the remaining two fell below the limit of detection (Fig 3E). These results confirmed that the nsp16.iORF sgmRNA is expressed during human infection.

The swabs we analysed were from three different lineages (B.1.1.44, B.1.416.1 and B.1.1.7/ Alpha), suggesting that the nsp16.iORF TRS may have been acquired convergently. Using Taxonium, we found at least 21 occurrences of convergent evolution of the nsp16.iORF TRS-B (Fig 3F and S1 Table), notably including the summer 2020 B.1.1.44 outbreak in Scotland (82% of sequences), a large Alpha subclade focussed on the Canadian Prairies (subsequently displaced by Delta) and a large AY.4.2 Delta subclade focussed on England (subsequently displaced by Omicron). As with the N.iORF3 TRS, we also observed further evolution of the sequence surrounding the nsp16.iORF TRS that increases the potential for base-pairing to the 5′UTR (C21303U, nsp16:P215L) (Fig 3B and 3C) and is present in 48% of sequenced genomes with the nsp16.iORF TRS. Two of our swab samples contained this additional mutation (Fig 3B and 3E), indicated by bullseye icons.

There are numerous other novel TRSs with lower prevalence in the global SARS-CoV-2 population. For example, the third-most common novel TRS has evolved independently at least seven times via a single nucleotide change within the connector domain [34] of Spike,

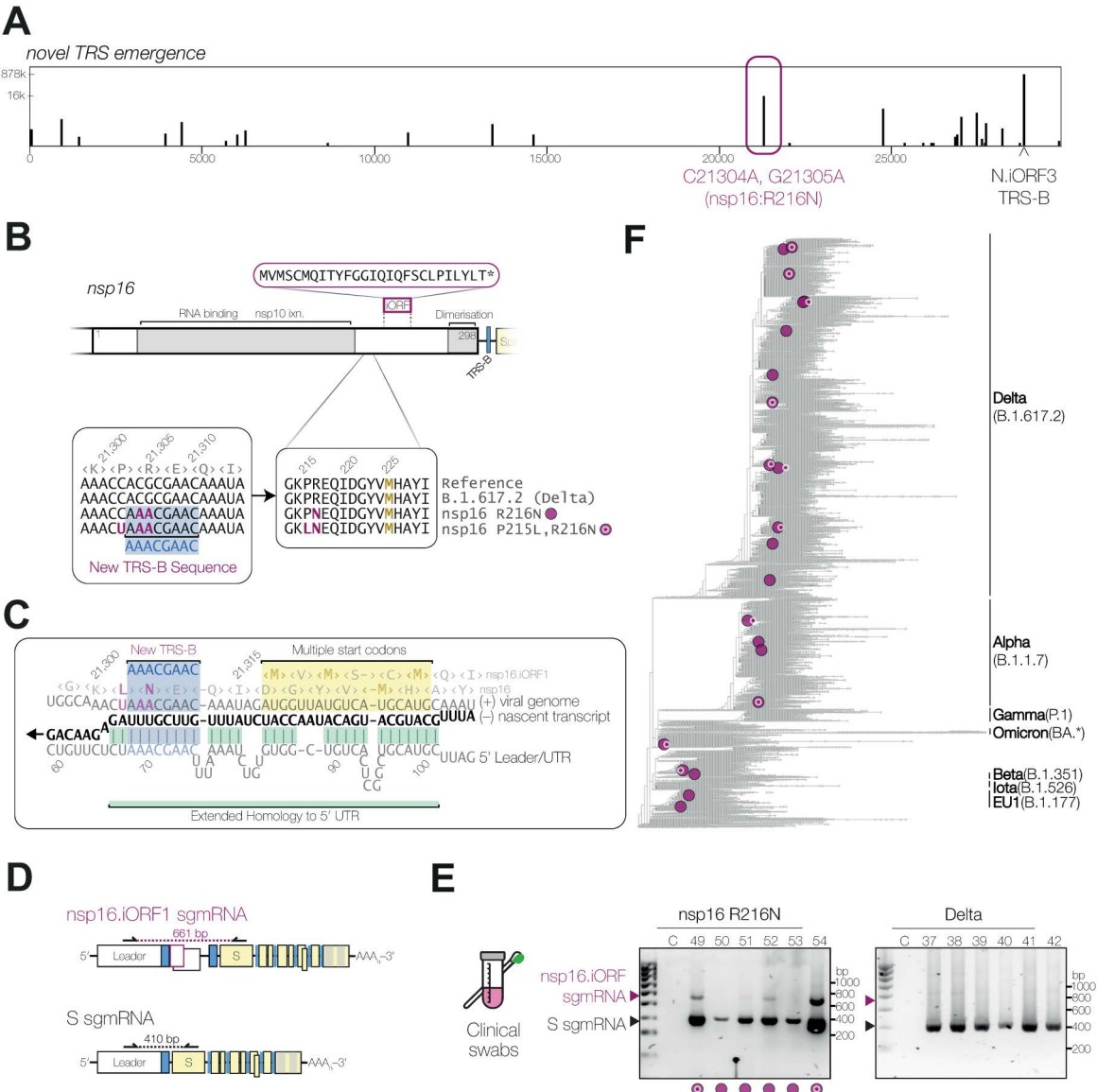

**Fig 3. Convergent evolution of a novel TRS-B within ORF1ab. (A)** Frequency of emergence of the TRS-B consensus sequence (AAACGAAC) in the global SARS-CoV-2 population. **(B)** Diagram of the nsp16 coding region, including a potential transframe product and sequence alignment of nucleotides 21298–21315, and amino acids 213–229 of nsp16, showing emergence of a new TRS-B sequence. **(C)** The sequence context of the novel nsp16.iORF sgmRNA, showing TRS-B (blue highlight), extended homology to the 5′UTR (green highlight) during nascent (−) strand RNA synthesis (black), and downstream start codon and Kozak context (yellow highlight). **(D)** Schematic representation of reverse transcription PCR (RT-PCR) analysis of RNA extracted from nasopharyngeal swabs, showing positions of primers. **(E)** RT-PCR detection of nsp16.iORF sgmRNA in clinical swabs (numbered 49–60), indicated by purple arrowheads, or Spike sgmRNA, indicated by black arrowheads. C, control PCR without template. **(F)** Phylogenetic reconstruction of SARS-CoV-2 evolution in humans, with independent emergences of nsp16.iORF TRS sequence with ≥100 descendant genomes highlighted in purple (see S1 Table). Emergence of extended homology to the 5′UTR is indicated with white outline "bullseye" pattern. Data underlying this figure can be found in: https://doi.org/10.25418/crick.27953013.

between the S1 and S2 subunits (S3 Fig and S1 Table), which contains extended homology to the 5′UTR and lies upstream of tandem out-of-frame start codons. Finally, subgenomic RNAs may also be produced from shorter or non-consensus TRSs [19]; one example is the canonical E sgmRNA, which is synthesised via a shorter TRS, ACGAAC, which we refer to here as the

minimal TRS (minTRS). The minTRS has emerged numerous times throughout the SARS-CoV-2 genome (S4A Fig), most notably >13 times at two loci in SARS-CoV-2 upstream of E (S4B Fig). This neo-minTRS lies within the coding region for the disordered CTD of ORF3a (S4C–S4E Fig) and came to be present in nearly all Australian Delta variant sequences prior to the arrival of the Omicron variant (S4F Fig).

Together, our analysis confirms the presence of at least two novel sgmRNAs which have evolved in different lineages of SARS-CoV-2 and highlights the prevalence of TRS-B evolution throughout the viral genome over the course of SARS-CoV-2 evolution to date.

## Expression and function of N.iORF3

Having confirmed the presence of non-canonical sgmRNAs from novel TRS-B sites, we then examined the N.iORF3 sgmRNA in more detail. Initially, we wanted to accurately quantify N.iORF3 expression during infection. Previous reports have suggested that N.iORF3 may be highly expressed in B.1.1-lineage infections, based on amplicon sequencing data [17,18]. Similarly, our RT-PCR experiments and subsequent nanopore sequencing showed that the N.iORF3-specific product accounted for over half of PCR amplicons, suggesting high expression (S1B Fig). However, short PCR products, such as the N.iORF3 amplicon, may be preferentially amplified, and thus overrepresented in these experiments [28]. Therefore, we designed a specific RT-qPCR assay to accurately compare N.iORF3 sgmRNA with other viral RNA species. Probes spanned the leader-TRS-sgmRNA junction (Fig 4A), allowing absolute quantitation of N.iORF3 sgmRNA copy number, compared to a cDNA standard (S5 Fig), as well as two canonical sgmRNAs: N, the most abundant viral transcript, and Envelope (E), whose expression is typically 100- to 1,000-fold lower than N [35]. These sgmRNAs therefore provide a useful reference frame to understand N.iORF3 expression levels within the context of established viral sgmRNAs.

RNA from human swabs, described above, was reanalysed by RT-qPCR: in Alpha- and Omicron-positive swab samples, N.iORF3 expression was comparable (30%–150%) to E sgmRNA expression (Fig 4B), while expression was approximately 100-fold lower than the highly abundant N sgmRNA (S6A Fig). We also analysed RNA from VeroE6 cells which were infected with a range of isolates both within and without the B.1.1 lineage, from different times during the COVID-19 pandemic. In addition to the Lineage B, Alpha and Beta isolates, described above, we infected VeroE6 cells with an early 2020 UK isolate (Lineage B.1.1), representative of a precursor virus to Alpha, and a late 2021 UK isolate (Omicron BA.1). Consistent with our clinical swab samples, in B.1.1, Alpha or Omicron infected cells, N.iORF3 expression was similar (60%–110%) to E expression (Fig 4C), and 50- to 100-fold lower than N sgmRNA (S6B Fig). These data show that amplicon-based sequencing data may overestimate N.iORF3 expression by up to 500-fold. We did not observe any substantial changes in E or N expression relative to genomic RNA levels at steady state, at either 7 or 24 h post-infection in any of the isolates examined (see S6 Fig).

We then examined the kinetics of sgmRNA expression during Alpha infection of a human lung carcinoma cell line, A549, which stably expresses the SARS2-CoV-2 receptor and entry co-factor ACE2 and TMPRSS2 (AAT). All sgmRNAs tested were expressed as early as 4 h post-infection and expression rapidly increased between 4 and 8 h post-infection, and started to plateau at 16–24 h post-infection (Fig 4D).

A previous report [17] suggested that the Alpha variant also expressed a distinct ORF9b-specific sgmRNA. While we were able to detect this product by nanopore sequencing of endpoint PCR products (S1B Fig), the expression level was below the limit of detection (100 copies) of our RT-qPCR assay using ORF9b-specific sgmRNA probes (S7 Fig), suggesting very low expression. Indeed, a recent report showed that increased ORF9b protein expression in

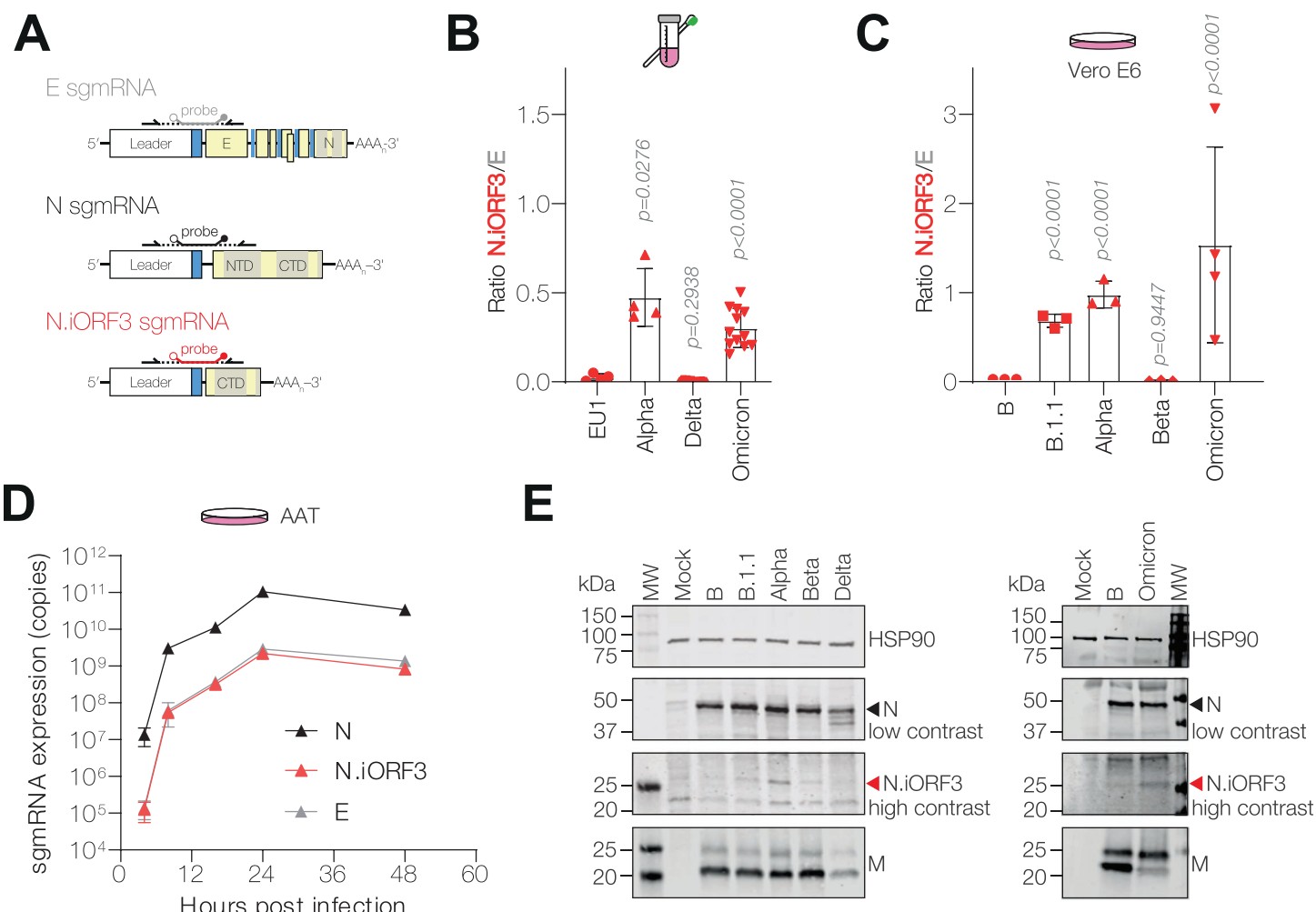

**Fig 4. N.iORF3 sgmRNA and protein are is expressed at low levels in infection. (A)** Schematic representation of reverse transcription qPCR (RT-qPCR) primer probe sets for Envelope (E), Nucleocapsid (N) and N.iORF3 sgmRNAs. **(B, C)** RT-qPCR analysis of N.iORF3 sgmRNA copy number, expressed as a ratio of E copy number in human clinical swabs (B) and infected cells (C). For clinical swabs, data are means and standard deviations of 4 (EU1/Alpha) or twelve (Delta/Omicron) swab samples per lineage, compared by one-way Brown-Forsythe and Welch ANOVA and Dunnett's T3 test, to account for unequal variances. For infected VeroE6 cells, data are means and standard deviations of at least three biological replicates, compared to 'Lineage B' or to Alpha by one-way ANOVA and Dunnett's test. *P* values are shown. **(D)** Dynamics of viral RNA expression during infection of Alpha in A549-ACE2-TMPRSS2 cells (AAT), showing absolute sgmRNA copy numbers. Data are means and standard deviations of three biological replicates. **(E)** Western blot analysis of lysates from infected VeroE6 ACE2-TMPRSS2 cells. N.iORF3 is indicated with a red arrowhead. MW, molecular weight marker. Data underlying this figure can be found in: https://doi.org/10.25418/crick.27952842 and https://doi.org/10.25418/crick.27953013.

Alpha and related viruses was due to enhanced leaky ribosomal scanning of the canonical N sgmRNA, rather than ORF9b sgmRNA expression [36].

## N.iORF3 protein is expressed in B.1.1 infections and functions as an innate immune antagonist

We next sought to determine whether a protein product was produced from N.iORF3 in infected cells. VeroE6 ACE2-TMPRSS2 cells were infected with Lineage B, B.1.1, Alpha, Beta or Delta viruses, and harvested at 20 h post-infection. When analysed by immunoblotting, N expression was largely consistent between variants. We also observed a band at ~25 kDa in B.1.1-, Alpha- and Omicron-infected cells (Figs 4E and S8A and B), consistent with the

predicted molecular weight of N.iORF3 protein. In line with our qPCR data, expression of the N.iORF3 band was very low compared to the main N band. We also observed a low level of N.iORF3 in Beta-infected cells, but not in Lineage B- or Delta-infected cells. We did not observe substantial N.iORF3 sgmRNA expression in Beta infection (see Fig 2F), therefore we speculate this protein product may arise via a separate mechanism, such as proteolytic processing [37].

To confirm the identity of this protein, we generated mutant viruses by reverse genetics to introduce or delete the N.iORF3 TRS-B (Fig 5A). Introducing the G28881A, G28882A, G28883C (N:R203K,G204R) mutations into a Wuhan-Hu-1_S:D614G virus background (WT-N:KR) increased the expression of N.iORF3 protein (S8C Fig), while reverting this

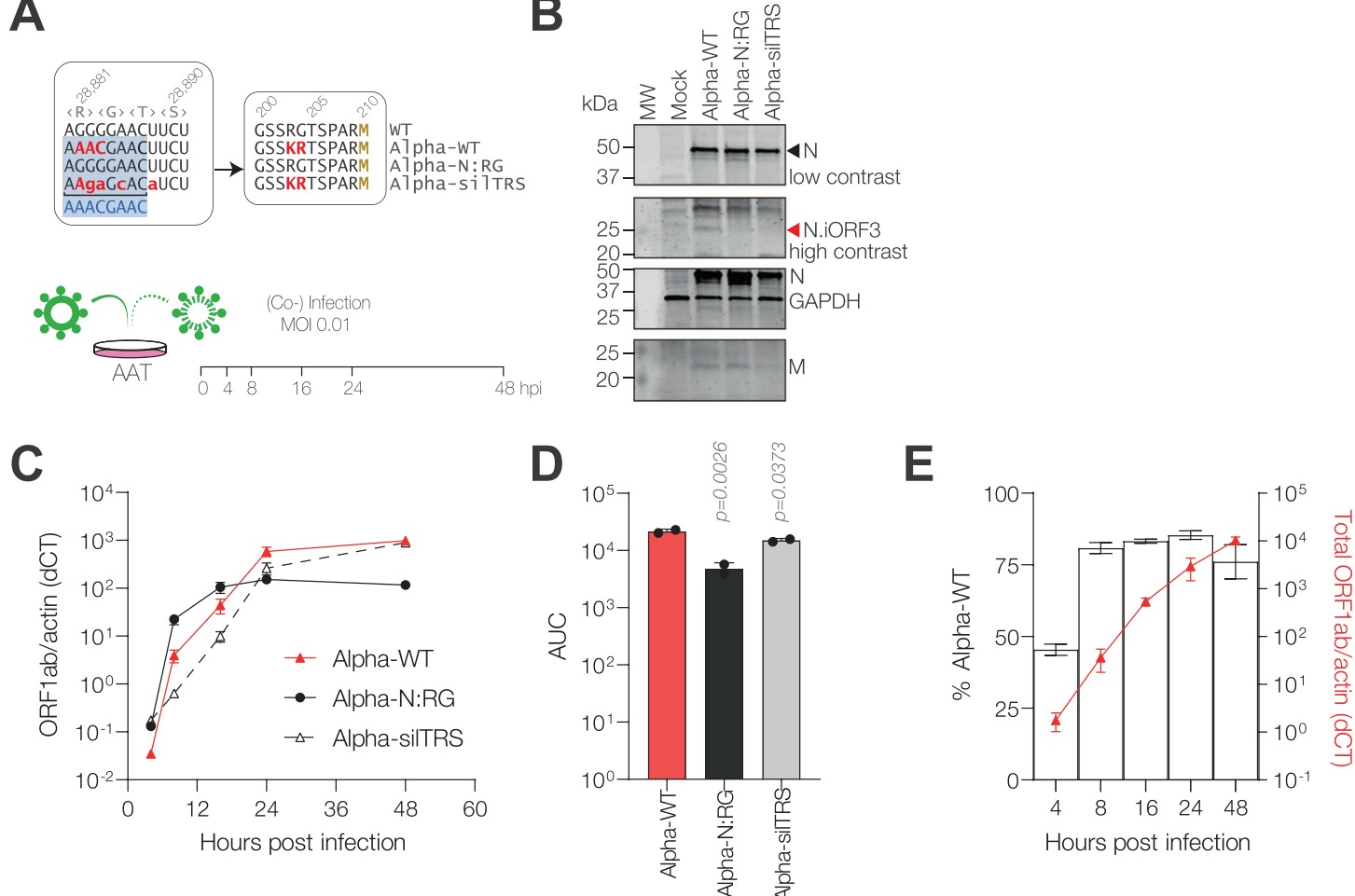

**Fig 5. N.iORF3 leads to the expression of a truncated form of Nucleocapsid and contributes to viral fitness. (A)** Nucleotide and amino acid sequences of reverse-genetics-derived SARS-CoV-2 mutant viruses (upper panel) and schematic of the experimental set-up for viral competition assays (lower panel). **(B)** Western blot analysis of lysates from infected VeroE6 cells. N.iORF3 is indicated with a red arrowhead. The membrane was re-probed for GAPDH after N, in the same fluorescent channel. MW, molecular weight marker. **(C)** Replication of reverse-genetics-derived viruses in A549-ACE2-TMPRSS2 (AAT) cells, measured by reverse transcription qPCR (RT-qPCR) against ORF1ab, normalised to actin. Data are means and standard errors of six biological replicates across two independent experiments. ANOVA analyses for individual times post-infection are given in S2 Table. **(D)** Corresponding area under the curve (AUC) values are means and standard deviations of AUC values from each independent experiment, compared to Alpha-WT infection by one-way ANOVA and Dunnett's test. P values are shown. **(E)** Head-to-head competition assays comparing fitness of Alpha-WT and Alpha-silTRS viruses, measured by Illumina sequencing of amplicons spanning the N.iORF3 TRS-B region and expressed as percentage of Alpha-WT reads. Total ORF1ab expression, normalised to actin, is shown on the right y-axis for reference. Data are means and standard deviations of three biological replicates. Data underlying this figure can be found in: https://doi.org/10.25418/crick.27952842 and https://doi.org/10.25418/crick.27953013.

mutation in an Alpha virus background (N:K203R,R204G; Alpha-N:RG) abolished expression of N.iORF3 protein (Fig 5B). We additionally made viruses in which the TRS-B was silently mutated, while maintaining the N amino acid sequence (WT-N:KR-silTRS and Alpha-silTRS; Fig 5A); alternative codons were chosen which had similar frequency to the Alpha-WT sequence. We found that introduction of the silTRS mutations also abolished N.iORF3 protein expression (Figs 5B and S8C), confirming that the TRS-B nucleotide sequence is necessary for expression of N.iORF3 protein. Both N.iORF3 TRS-B mutations resulted in a drastic decrease in N.iORF3 sgmRNA expression, as determined by RT-qPCR (S9 Fig); we also noticed small differences in ORF4 expression, with a decrease relative to Alpha-WT in Alpha-N:RG viruses and an increase in Alpha-silTRS (S9B Fig), but no significant changes were observed for N sgmRNA expression.

To determine the fitness of these viruses in cell culture, AAT cells were infected at a low multiplicity and cells were harvested up to 48 h post-infection for RT-qPCR analysis. Introduction of the N:KR or N:KR-silTRS mutations into a WT background did not substantially affect viral replication until 24 h post-infection, when the WT-N:KR virus grew to slightly higher peak levels compared to WT (2.8-fold, $p = 0.0292$, one-way ANOVA with Tukey's multiple comparisons test) and WT-N:KR-silTRS (3.4-fold, $p = 0.0092$) (S10A–S10C Fig). We then analysed the Alpha backbone viruses. The Alpha-N:RG mutant, which contains the WT-like amino acid and nucleotide sequence, was attenuated relative to Alpha, with reduced overall replication and lower peak genome expression at 24–48 h post-infection (8.4-fold at 48 hpi, $p < 0.0001$), despite slightly higher genome copies at early times post-infection (Fig 5C and 5D). Conversely, the Alpha-silTRS mutant showed substantially delayed growth compared to Alpha-WT up to 16 h post-infection, which recovered by 48 h post-infection (Fig 5C and 5D and S2 Table).

We then tested these viruses in head-to-head competition assays: AAT cells were infected with pairs of viruses at a low multiplicity and harvested over 48 h. Total virus replication was quantified by RT-qPCR while relative fitness was determined by amplicon sequencing of the region surrounding the N.iORF3 TRS-B, expressed as the percentage of sequencing reads mapping to each virus. The proportion of Alpha-WT virus increased over time relative to Alpha-N:RG, confirming a selective advantage for Alpha-WT (S10D Fig). Likewise, Alpha-WT rapidly out-competed the Alpha-silTRS mutant, reaching >80% of amplicon reads by 8 h post-infection (Fig 5E). Since the growth kinetics of the Alpha-N:RG and Alpha-silTRS mutants were distinct, we also tested these viruses against each other in a head-to-head competition assay. Alpha-N:RG dominated between 4 and 16 h post-infection, consistent with its faster growth kinetics at early time points, but Alpha-silTRS overtook by 24–48 h, reaching 75% of amplicon reads by the end of the time course (S10E Fig). Together these results indicate that both the amino acid and the nucleotide mutations contribute to fitness in the context of the Alpha backbone.

Finally, we investigated how N.iORF3 may contribute to viral fitness. In SARS-CoV, type I interferon (IFN) production is strongly inhibited by N, specifically its C-terminal domain, which sequesters double-stranded RNA in the cytoplasm away from host pattern recognition receptors [38,39]. We therefore hypothesised that N.iORF3 protein, which encompasses the CTD of N, might act as an antagonist of type I IFN induction. We therefore investigated whether knocking out either of the major dsRNA sensors in the cytoplasm, MDA5 or RIG-I, could restore fitness in viruses which lack N.iORF3. We infected A549-dual ACE2-TMPRSS2 (WT), or corresponding MDA5 knockout (MDA5 KO) and RIG-I KO cells with a low multiplicity of Alpha-WT, Alpha-N:RG and Alpha-silTRS, and examined viral replication over 48 h by RT-qPCR (Fig 6A). Knockout of MDA5 did not have a significant impact on replication of Alpha-WT or Alpha-silTRS viruses (S11A and S11C Fig), while Alpha-N:RG genome levels were slightly increased at late times post-infection (S11B Fig).

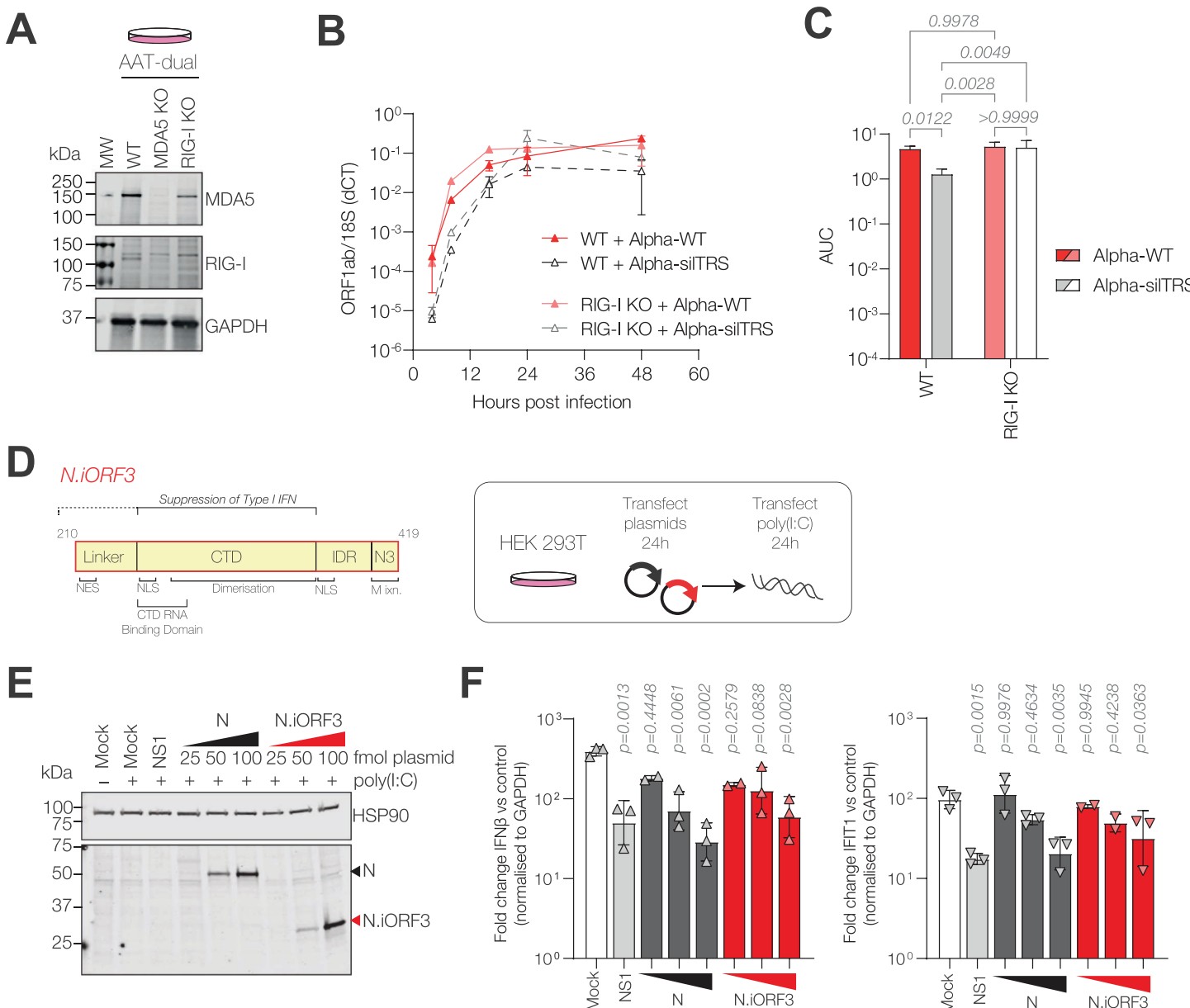

**Fig 6. N.iORF3 antagonises interferon induction downstream of RIG-I. (A)** Western blot analysis of lysates from AAT-dual WT, MDA5 KO and RIG-I KO cell lines. **(B)** Replication of reverse-genetics-derived viruses in WT and RIG-I KO AAT-dual cells, measured by reverse transcription qPCR (RT-qPCR) against ORF1ab, normalised to 18S rRNA. Data are means and standard deviations of three biological replicates. ANOVA analyses for individual times post-infection are given in S3 Table. **(C)** Corresponding area under the curve (AUC) values, compared by two-way ANOVA with Tukey's multiple comparisons. *P*-values are shown. **(D)** Diagram of N.iORF3 protein, also called N *, showing domains from Nucleocapsid and described functions (left panel) and schematic showing experimental design (right panel). **(E)** Western blot analysis of N and N.iORF3 expression in transfected HEK293T cells. **(F)** Expression of IFNb (left panel) and a representative interferon-stimulated gene (IFIT1, right panel), normalised to GAPDH and expressed as fold change in cells transfected with poly(I:C) compared to control cells which were not transfected with poly(I:C), in the presence of increasing concentrations of N- or N.iORF3-expressing plasmids (25, 50 or 100 fmol) or NS1 from influenza A virus as a positive control (100 fmol). Data are means and standard deviations of at least two biological replicates, compared to mock-transfected cells by one-way ANOVA and Dunnett's test. *P*-values are shown. Data underlying this figure can be found in: https://doi.org/10.25418/crick.27952842 and https://doi.org/10.25418/crick.27953013.

In RIG-I KO cells, growth of Alpha-WT was slightly increased at 8 and 16 h post-infection (Figs 6B, S11A and S11D), though overall replication, as determined by area under the curve (AUC) analysis, was comparable between cell lines (Fig 6C, *p* = 0.9978). This replication

advantage was more pronounced in Alpha-silTRS, with significantly higher overall replication compared to WT cells (3.9-fold, $p$ = 0.0049, two-way ANOVA with Tukey's multiple comparisons test) which was comparable to Alpha-WT ($p$ > 0.9999) (Fig 6C). Replication of Alpha-silTRS was still delayed relative to Alpha-WT, but recovered by 24 h post-infection (Fig 6B). Likewise, Alpha-N:RG replicated to higher levels in RIG-I KO cells compared to WT cells, with higher peak genome levels at 16–48 h post-infection (S11B Fig), though overall replication was not significantly restored (S11D Fig, $p$ = 0.5336). Together these data show that removal of RIG-I can partially compensate for the loss of N.iORF3, indicating that N.iORF3 acts as a RIG-I antagonist.

To test this directly, HEK293T cells were transfected with plasmids encoding N or N.iORF3, or influenza A virus NS1 protein as a positive control, since NS1 is a potent type I IFN antagonist [40]; after 24 h, cells were transfected with poly(I:C), a synthetic double-stranded RNA analogue which stimulates innate immune signalling following sensing by RIG-I or MDA5. Cells transfected with either N or N.iORF3 expressed lower mRNA levels of type I IFN (IFNβ) and a representative IFN-stimulated gene (IFIT1) (Fig 6F), indicating that N.iORF3 can antagonise IFN signalling downstream of dsRNA sensing in the cytoplasm.

## Discussion

Since its emergence in late 2019, SARS-CoV-2 has continued to adapt to the human host. Many of these adaptations have been within Spike, the major glycoprotein on the virion surface, to increase affinity for its receptor, ACE2, or to evade recognition by the adaptive immune system [2–4]. However, there have also been numerous changes to genes outside of Spike and in non-coding regions [5,41]. Here, we have focussed on the emergence of novel TRSs during SARS-CoV-2 evolution. We have shown that TRS emergence happens frequently on a global scale, may occur convergently across different lineages, and can lead to the expression of novel subgenomic RNAs which encode novel protein isoforms.

In SARS-CoV-2, the most common newly-emerged TRS-B occurs within the Nucleocapsid gene, at nucleotides 28881–28883 (AGGGGAAC → AAACGAAC), underlying the B.1.1 lineage-defining mutations N:R203K,G204R. We confirmed that these nucleotide mutations lead to the expression of a new subgenomic ORF, N.iORF3, and using a custom sgmRNA-specific qPCR assay, we accurately quantified N.iORF3 expression in both cell culture and swab samples from infected individuals. We found that amplicon sequencing data overrepresented N.iORF3 expression (S1 Fig and [16–18]), likely due to PCR bias of the shorter N.iORF3 amplicons [28]. This highlights the complexity of accurately quantifying overlapping Nidovirus RNA species using indirect sequencing methods [42]. Of note, we have extended the utility of our sgmRNA qPCR probe design to allow sgmRNA-specific knockdown in the context of infection, allowing for mutational analysis of the Envelope protein without the need for reverse genetics [43].

We examined the contribution of this novel sgmRNA to viral fitness. Previous reports have shown that viruses bearing the N:R203K,G204R mutation have an advantage in cell culture and in an in vivo hamster infection model over WT virus [44,45]. These studies focussed on the coding consequences of the N:KR mutation, which occurs in the middle of the serine-arginine-rich linker in between the structured N-terminal and C-terminal domains of N (Fig 2C). This linker is phosphorylated during infection [46], and regulates the RNA chaperone [47,48] and genome packaging roles of N [49]. N:KR was shown to increase the phosphorylation of this linker [36,44] and alter the dynamics of N phase separation in vitro [50], suggesting that the amino acid substitutions may alter N regulation during infection. Structural modelling has also suggested that N RNA binding activity may be altered by the slight alteration in charge distribution within this domain [45].

Here, we confirmed the fitness advantage of N:KR-bearing viruses in IFN-competent cells, particularly in the context of an Alpha-lineage virus backbone, but add that the advantage is diminished when the nucleotide sequence underlying the N:KR mutation is mutated to ablate the TRS-B (silent TRS mutant, silTRS). Interestingly, we observed distinct attenuation phenotypes in these mutant viruses: replication of Alpha-silTRS was drastically delayed, but recovered to wild-type Alpha levels by 48 h post-infection. By contrast, mutation of the N coding sequence to N:RG, which also ablates N.iORF3 expression, resulted in lower peak replication at late times post-infection, even though replication was similar to wild-type Alpha at early time points. Likewise, in direct competition assays, the Alpha-N:RG virus had a fitness advantage over Alpha-silTRS at early times post-infection, but at later times Alpha-silTRS recovered to become the dominant virus in the population. We therefore conclude that the N:R203K,G204R mutation is advantageous because of both the coding and non-coding changes, and add that there appears to be a crosstalk between the nucleotide and amino acid sequence changes which warrants further investigation.

We further demonstrated that N.iORF3 protein can act as an inhibitor of IFN induction, contributing to the panel of innate immune antagonists which SARS-CoV-2 has at its disposal, providing a potential mechanism the observed fitness advantage [18]. We found that knockout of RIG-I, but not MDA5, in A549 cells could restore replication of Alpha-silTRS back to Alpha-WT levels over the course of infection, indicating that N.iORF3 acts as a RIG-I antagonist. RIG-I has been shown to bind to the 3′UTR of the SARS-CoV-2 genome and was shown to be the primary mediator of innate immune induction in both primary and A549 lung epithelial cells [51–53]; this may vary in a cell type-dependent manner [54,55], with other lung epithelial cell lines such as Calu-3, showing equal or dominant MDA5 sensing and restriction of SARS-CoV-2 infection [56–58]. It is plausible that the C-terminal RNA binding domain of N.iORF3 acts as a competitive inhibitor of RIG-I binding, in an analogous manner to the C-terminal domain of SARS-CoV nucleocapsid [38,39,59]. While N.iORF3 protein expression is low compared to N, the uncoupling of the N- and C-terminal domains may relieve N.iORF3 of some of the other roles of N during infection, such as RNA chaperoning during replication and transcription, which is mediated by the N-terminal domain. Additionally, N.iORF3 was recently reported to bind to the RNA polymerase II-associated factor complex (PAFc) [36], similar to Influenza virus NS1 protein, which potently downregulates host innate immune responses by reprograming host transcription [60], offering another potential mode of action for innate immune antagonism, which is the subject of ongoing investigation.

However, we found that while peak replication of Alpha-silTRS was significantly increased in the absence of RIG-I, replication still lagged behind Alpha-WT virus, indicating incomplete restoration of WT-like replication. This indicates that there may be multiple roles for N.iORF3 during infection beyond RIG-I antagonism. Indeed, N.iORF3 has been shown to form ribonucleoprotein complexes in a similar manner to full-length N [61] and could mediate viral RNA packaging in the absence of N in a virus-like particle assay [36], indicating a potential role in virus assembly. N.iORF3 may also modulate the function of full-length N: the CTD of N, and consequently N.iORF3, contains a dimerisation domain, raising the possibility of N:N.iORF3 heterodimers. Intriguingly, we identified another internal ORF downstream of a neo-TRS-B at the 3′ end of ORF1ab, which encodes two ORFs, one of which encompasses the C-terminal tail of nsp16, the viral RNA 2′-O-methyltransferase [62]. This region includes the dimerisation domain of nsp16, similarly suggesting heterodimerisation between the full-length and truncated proteins arising from these newly evolved TRS-Bs.

The generation of new open reading frames is a general feature of coronavirus evolution [63]. This is often considered in the context of dramatic duplications or rearrangements which generate new ORFs which are then able to diversify in function; examples in SARS-CoV-2

include the ORF3a, an accessory protein which evolved from a copy of the viral Membrane protein [64], and ORF8 which is thought to be a duplicate of ORF7a [65–67], both accessory proteins with diverse functions in host cell modulation. Functional redundancy between the Membrane and Envelope genes of mouse coronavirus even suggests a common evolutionary origin of these two structural proteins in early coronavirus evolution [68]. Moreover, novel ORFs may arise via non-homologous recombination with unrelated viruses: an ancestral betacoronavirus is thought to have acquired its haemagglutinin esterase gene from an influenza C-like virus [69,70], while a bat coronavirus has been identified with a novel 3′ ORF which was homologous to a gene from a bat reovirus, a double-stranded RNA virus [71]. These sequences can then be exchanged or shuffled between viruses to provide rapid sequence diversity. Notably, regions of the Spike gene, described as mobile "modules", can be exchanged between similar viruses and have been implicated in expansion of viral host range [72,73]. In SARS-CoV-2, recombination between two subvariants of Omicron, BJ.1 and BM.1.1.1, within the Spike gene lead to the generation of the XBB variant, which dominated globally throughout 2023 [74,75].

Here, we have shown that ORF evolution can also occur from more subtle mutations, even down to single nucleotide changes, which generate consensus TRS-Bs that are sufficient to drive novel sgmRNA synthesis. In the case of N.iORF3, we observed low-level expression in viruses without the TRS-B mutations, but expression was increased by at least 100-fold when the TRS-B mutations were present. This is in line with numerous previous reports which show that coronaviruses can recombine promiscuously with TRS-B-like sequences at low frequency, producing a wide diversity of subgenomic RNA species during infection beyond the canonical sgmRNAs [19–21], and artificial insertion of a consensus TRS alone is sufficient to enhance sgmRNA expression by 100- to 1,000-fold [76].

It is important to note that TRS usage varies greatly between different Nidoviruses [23,77]. For example, the TRS-Bs upstream of the HE, ORF5 (also known as ns12.9) and M in human coronavirus OC43 contain deletions or mutations relative to the TRS-L. In Sarbecoviruses, as noted above, transcription of the Envelope sgmRNA is driven by a shorter hexanucleotide TRS-B, while in OC43, TRS-B-like sequences upstream of E are both shorter and mutated relative to the TRS-L. A short TRS-B was also implicated in the transcription of a novel sgmRNA transcript in SARS-CoV, which encodes a truncated version of Spike [78]. Likewise, in avian coronavirus and other gammacoronaviruses, shorter TRS-Bs can be used highly efficiently [27]; indeed, a minimal TRS of only three nucleotides is efficiently used for the expression of gene 4b [79]. Moreover, in mouse coronavirus, reverse genetics experiments have revealed additional non-TRS-B sequences which can be used for sgRNA transcription [80]. Finally, in some arenaviruses, sgmRNA synthesis is even more plastic and can be both discontinuous and non-discontinuous (terminating without addition of the 5′ leader) [81–83].

There are numerous other sequence and structural features which govern TRS-B selection [84,85], including the identity of TRS-flanking nucleotides [11,13,14], the structural presentation of the TRS [86] and long-range RNA–RNA interactions between the 5′ and 3′ ends of the genome [12,15]. Position in the genome is also a key factor, with more 3′ TRS-Bs being selected more frequently than those 5′ [87]. Indeed, in both swine and mouse coronaviruses, experimental introduction of optimal TRS-Bs downstream of N leads to strong expression of the new transgene, and downregulation of the upstream ORFs [76,88]. By contrast, we found that N.iORF3 expression was ~100-fold lower than N, at levels comparable to the E sgmRNA. Additionally, we observed no differences in canonical sgmRNA expression in the presence or absence of the N.iORF3 TRS-B, indicating minimal perturbation of the overall transcriptional programme. We may conclude that the N.iORF3 TRS-B is inefficiently utilised, and may have been compensated for with other mutations in the B.1.1 lineage, possibly to

avoid dysregulated sgmRNA transcription which could be deleterious [18,87]. However, we did observe small changes to E sgmRNA expression in our reverse-genetics-derived viruses, indicating that when the TRS-B is perturbed artificially, the transcriptional programme may be altered. This therefore implies that other changes are present in viruses that contain the N.iORF3 TRS-B, which could compensate for the additional TRS; additional reverse genetics experiments may shed light on this potential transcriptional fine-tuning in the future. It is tempting to speculate that inefficiently expressed transcripts may provide an opportunity for new ORFs to be expressed and, if found advantageous, the efficiency of the TRS-B may then be further optimised.

In summary, the emergence of novel open reading frames is a hallmark of coronavirus evolution and adaptation to new hosts in coronaviruses [64,72,89], Nidoviruses [63,90,91], RNA viruses more broadly [92]. Indeed, gene duplication and diversification as an evolutionary process is fundamental to complexity in cellular organisms and forms the backbone of both innate and adaptive immune systems. Here, we show that this process is ongoing in SARS-CoV-2, as we observe evolution at the functional RNA level throughout the genome.

## Materials and methods

### Cell lines and viruses

Vero E6 (Pasteur), Vero V1 (a gift from Stephen Goodbourn), and A549 ACE-TMPRSS2 and VeroE6 ACE2-TMPRSS2 cells [93] (gifts from Suzannah Rihn) were maintained in Dulbecco's Modified Eagle Medium (DMEM), supplemented with 10% foetal calf serum and penicillin-streptomycin (100 U/mL each). A549-dual hACE2-TMPRSS2, A549-dual KO MDA5 hACE2-TMPRSS2 and A549-dual KO RIG-I hACE2-TMPRSS2 cells were purchased from Invivogen, and were maintained in DMEM as above, with addition of 100 μg/mL Normicin, 10 μg/mL Blasticidin, 100 μg/mL Hygromycin, 0.5 μg/mL Puromycin and 100 μg/mL Zeocin. Forty-eight hours before infection selection antibiotics were removed to avoid off-target pressure on viral growth.

The SARS-CoV-2 B lineage isolate used (hCoV-19/England/02/2020) was obtained from the Respiratory Virus Unit, Public Health England, UK, (GISAID accession EPI_ISL_407073). The B.1.1 lineage strain used was isolated from a healthcare worker swab as part of the Legacy study and has the genotype: C241T, C3037T, nsp12: P323L, S: D614G, N: S194L, N: R203K, N: G204R. The SARS-CoV-2 B.1.1.7 isolate ("Alpha") was hCoV-19/England/204690005/2020, which carries the D614G, Δ69-70, Δ144, N501Y, A570D, P681H, T716I, S982A and D1118H mutations in Spike, and was obtained from Public Health England (PHE), UK, through Prof. Wendy Barclay, Imperial College London, London, UK through the Genotype-to-Phenotype National Virology Consortium (G2P-UK). The B.1.617.2 ("Delta") isolate was MS066352H (GISAID accession number EPI_ISL_1731019), which carries the T19R, K77R, G142D, Δ156-157/R158G, A222V, L452R, T478K, D614G, P681R, D950N mutations in Spike, and was kindly provided by Prof. Wendy Barclay, Imperial College London, London, UK through the Genotype-to-Phenotype National Virology Consortium (G2P-UK). The BA.1 ("Omicron") isolate was M21021166, which carries the A67V, Δ69-70, T95I, Δ142-144, Y145D, Δ211, L212I, G339D, S371L, S373P, S375F, K417N, N440K, G446S, S477N, T478K, E484A, Q493R, G496S, Q498R, N501Y, Y505H, T547K, D614G, H655Y, N679K, P681H, A701V, N764K, D796Y, N856K, Q954H, N969K and L981F mutations in Spike, and was kindly provided by Prof. Gavin Screaton, University of Oxford, Oxford, UK through the Genotype-to-Phenotype National Virology Consortium (G2P-UK). The B.1.351 isolate ("Beta") was obtained from Alex Sigal and Tulio de Olivera. Viral genome sequencing of this B.1.351 identified S: Q677H and S: R682W mutations at the furin cleavage site in ∼45% of genomes.

Reverse-genetics-derived viruses were generated at the CVR, using the transformation-associated recombination method, as previously described [94,95]. Genomes were transcribed in vitro and transfected into BKH-hACE2-N cells, which stably express hACE2 and SARS-CoV-2 N, for virus rescue. Rescued viruses were passaged twice in Vero E6 cells and sequenced using Oxford Nanopore to confirm their identity. Virus stocks were propagated in Vero V1 cells by infection at an MOI of 0.01 in DMEM, supplemented with 1% foetal calf serum and penicillin-streptomycin (100 U/mL each), harvested when CPE was visible, and stocks were titrated on Vero E6 cells.

## Clinical samples

Extracted RNA from occupational health screening swab samples of UK National Health Service (NHS) healthcare workers at the Crick COVID Consortium Testing Centre18 was obtained from the Crick/UCLH SARS-CoV-2 Longitudinal Study (Legacy Study) [COVID-19] (IRAS ID 286469). These samples were collected between December 2020 and February 2021 and had tested positive for SARS-CoV-2 (TaqPath assay, Thermo Fisher) and the viral genomes had been fully sequenced (ARTIC v333,34, GridION), and lineage assigned using pangolin [1]. The 12 samples with the lowest ORF1ab Ct values and a genome coverage >96% were selected from each of Alpha (B.1.1.7) and B.1.177 (EU1) lineage for use in this work.

## Ethics

The Legacy study (Cohort A1, NCT04750356) was approved by London Camden and Kings Cross Health Research Authority Research and Ethics committee (IRAS number 286469) and is sponsored by University College London Hospitals. Participants provided written consent.

## Plasmids

Sequences for N and N.iORF3 were amplified from cDNA from Alpha-infected cells, to include 5′ NheI and 3′ NotI sites, then ligated into pCDNA3-T2A-mCherry, to generate pCDNA3-B117-N-T2A-mCherry and pCDNA3-B117-Nstar-T2A-mCherry. pCDNA3-NS1 was a kind gift from Caetano Reis e Sousa.

## In vitro infections

For in vitro infections to examine sgmRNA production, Vero E6 cells were infected at an MOI >1 in DMEM supplemented with 1% foetal calf serum and penicillin streptomycin (100 U/mL each). At 7 or 24 h post-infection, cells were washed with phosphate-buffered saline and lysed in TRIzol or Laemmli buffer. For growth curves and competition assays, A549 ACE2 TMPRSS2 (AAT) cells, A549-dual hACE2-TMPRSS2, A549-dual KO MDA5 hACE2-TMPRSS2 or A549-dual KO RIG-I hACE2-TMPRSS2 cells were infected at an MOI of 0.01 PFU/cell at room temperature for 1 h. For competition assays, cells were infected with both viruses at an MOI of 0.01 each. Inoculum was then removed and replaced with fresh DMEM, supplemented with 2% foetal calf serum and penicillin streptomycin (100 U/mL each). Cells were incubated for 4–48 h, before lysis in Trizol. RNA from infected cells was extracted using a Direct-zol RNA MiniPrep kit (Zymo Research).

## RT-PCR

Reverse transcription was carried out using the SuperScript VILO cDNA synthesis kit (Invitrogen) at 25 °C for 10 min, then 50 °C for 10 min, followed by 2 min each at 55, 50, 60, 50, 65, and 50 °C. PCR was carried out using GoTaq DNA polymerase (Promega) with RT-PCR

primers 5′UTR-FWD and ORF9-REV (S4 Table). PCR cycling conditions were 95 °C for 2 min, followed by 40 cycles of 95 °C for 30 s, 56 °C for 30 s, and 72 °C for 60 s, with a final extension of 72 °C for 5 min. PCR products were purified using a NucleoSpin Gel and PCR Clean-up kit (Macherey-Nagel). PCR products were separated in 1.2% agarose-TBE gels at 110 V for 45 minutes, before visualisation using an Amersham Imager 600 or Li-Cor D-Digit.

## Nanopore sequencing and data analysis

End preparation was carried out using the NEBNext Ultra II End Repair A-Tailing Module (NEB). The reaction was carried out at 20 °C for 25 min, followed by 15 min at 65 °C to inactivate the enzyme. Samples were directly ligated with an individual barcode from the ONT Native Barcoding kit (EXP-NBD196) using NEBNext Blunt/TA Ligase Master Mix (NEB). The ligation reaction was carried out for 20 min at 20 °C followed by 10 min at 70 °C. After barcoding, samples were pooled with equal volumes. The pool is cleaned up using a 0.4× ratio of SPRIselect magnetic beads (Beckman Coulter). Fragment-bound beads were washed twice with short fragment buffer (SFB) (ONT), and once with 80% ethanol. The DNA target was eluted with Buffer EB (Qiagen). To allow nanopore sequencing, the AMII adapter (ONT) was ligated on using NEBNext Quick Ligation Module (NEB), with a 20-min incubation at 20 °C. A second SPRIselect bead cleanup was carried out with a 100d7 ratio of beads and two washes with SFB only. The final pool was eluted in Elution Buffer (ONT) and quantified by Qubit HS dsDNA assay (Thermo Fisher). A FLO-MIN106 flowcell is primed with the Flow cell Priming kit (ONT). Up to 15 ng of the final pool was mixed with Loading Beads (ONT) and Sequencing Buffer (ONT) before loading onto the flowcell, which was sequenced on the GridION platform for up to 20 h with a voltage of −180 mV. Sequencing reads were adapter-trimmed using porechop (https://github.com/rrwick/Porechop) and reads were filtered by size into bins of 75–130 bp, 250–400 bp and 850–1,050 bp, and aligned to the reference sequence of the ORF9 amplicon using minimap2 [96] and samtools [97]. Alignments were visualised in Integrative Genomics Viewer [98], and statistics were extracted using the samtools flagstat command. ORF9b-specific amplicons in were identified by counting the number of reads in the 850–1,050 bin containing the ORF9b sgmRNA-specific sequence TGTAGATCT-GTTCTCTAAATGGACC. A consensus sequence was generated from reads in 75–130 bp bin, which identified this product as a result of mis-priming of 4 bases at the 3′-end of the reverse PCR primer. Quantification of read statistics were plotted using GraphPad Prism (v 9.2.0). Raw nanopore sequencing data has been deposited under the ArrayExpress accession number E-MTAB-14681.

## Illumina amplicon sequencing and data analysis

To determine the abundance of each virus in head-to-head competition assays, a region surrounding the N.iORF3 TRS-B sequence was amplified using Platinum SuperFi II DNA Polymerase (Invitrogen) with primers N3geno_FWD and N3geno_REV which contained 5′ Illumina adapter sequences. PCR cycling conditions were 95 °C for 5 min, followed by 20 cycles of 95 °C for 30 s, 60 °C for 30 s and 72 °C for 20 s, with a final extension of 72 °C for 5 min. Products were cleaned up using Ampure XP beads (Beckman Coulter) at a ratio of 1.8×. One microlitre of each sample was taken into a PCR reaction with 5 μL of NEB Q5 High-Fidelity DNA Polymerase 2× Master Mix, 1 μL of NXT-IDT Primer Mix 10 μM unique dual index and 3 μL of $H_2O$. PCR cycling conditions were 95 °C for 3 min, followed by 10 cycles of 95 °C for 30 s, 55 °C for 15 s, and 72 °C for 30 s, with a final extension of 72 °C for 5 min. Libraries were cleaned up using Ampure XP beads at a ratio of 0.8×. Libraries were quantified using Promega Quantifluor reagents, plate reader and Agilent Tapestation. Samples

were pooled by concentration and sequenced on the Illumina MiSeq platform with a PE 250 bp run configuration on a nano flowcell. Reads were aligned to reference sequences using Bowtie2 [99] and read counts were calculated using SAMtools [97], using a custom pipeline (https://doi.org/10.5281/zenodo.14277568). Amplicon data are presented as a percentage of the total number of reads which map to the indicated virus. Data were analysed by one-way ANOVA and tested for a linear trend using GraphPad (v. 10.1.2). Raw Illumina sequencing data has been deposited under the ArrayExpress accession number E-MTAB-14680.

### RT-qPCR

To quantify sgmRNA abundance, RT-qPCR was carried out using the Taqman Multiplex Master Mix (Applied biosystems) with 1.8 μM forward and reverse primers and μM 5′-FAM-/3′-BHQ-labelled probe and cDNA from 5 ng of RNA extracted from infected VeroE6 cells (or a 1:100 dilution for N), or from 2 μL of RNA from HCW swab samples. Linear amplification range was tested against synthetic oligonucleotide templates (Dharmacon) and absolute sgmRNA copy number was calculated by interpolating cDNA standard curves. Ratios between sgmRNA species were calculated from copy numbers. ORF1ab and Actin abundances were determined using the Real-Time Fluorescent RT-PCR kit for Detecting nCoV-19 (BGI), or, for S9B–D Fig, TaqPath COVID-19 Combo Kit (ThermoFIsher) and Taqman beta-actin detection Reagents (Applied Biosystems). For Fig 4, $\log_{10}$-transformed data from infected cells were compared to B Lineage-infected cells by one-way ANOVA with Dunnett's test for multiple comparisons using GraphPad Prism (v 9.2.0). To account for differences in variances due to uneven samples sizes from swab samples, data were compared to EU1 swabs by one-way Browne-Forsythe and Welch ANOVA with Dunnett's T3 test. For Figs 5, 6, S9 and S10, AUC values were calculated in GraphPad Prism (v 10.1.2) and compared by one-way ANOVA with Dunnett's test for multiple comparisons (Figs 5D and S9) or two-way ANOVA with Tukey multiple comparisons correction. Additionally, ANOVA comparisons were made for $\log_{10}$-transformed data from each time point and are listed in S2 and S3 Tables.

To determine innate immune responses, RT-qPCR was carried out using the Brilliant III Ultra-Fast SYBR Green QPCR Master Mix (Agilent) with 0.8 μM primers, against IFNb, IFIT1 or GAPDH (see "Primer Sequences", below), and cDNA from 5 ng of RNA extracted from transfected HEK293T cells. Data were normalised to GAPDH and expressed as fold change (2-DDCq) over control cells, which were not transfected with poly(I:C), for each plasmid. $\log_{10}$-transformed data were compared by one-way ANOVA with Dunnett's test for multiple comparisons using GraphPad Prism (v 9.2.0).

### Immunoblotting

Cell lysates (6–10 μg total protein) were separated by SDS-PAGE using Any kD precast gels (Bio-Rad) and transferred to 0.2-μm nitrocellulose membrane by semidry electrotransfer. Membranes were probed with anti-Nucleocapsid (MA5-35943, Invitrogen, 1:250), anti-HSP90 (MA5-35624, Invitrogen, 1:2000), anti-GAPDH (AM4300, ThermoFisher, 1:2000), anti-Membrane (MRC PPU & CVR Coronavirus Toolkit, Sheep No. DA107, 1:800), anti-MDA5 (XX) or anti RIG-I (XX), followed by IRdye secondary antibodies (Li-Cor), allowing visualisation on an Odyssey CLx imaging system (Li-Cor).

### Transfection

HEK293T cells were transfected at 70% confluency in 24-well plates, with 25, 50 or 100 fmol pCDNA3-B117-N-T2A-mCherry or pCDNA3-B117-Nstar-T2A-mCherry from Alpha (B.1.1.7), or 100 fmol pCDNA-NS1, or mock transfected, using Lipofectamine 2000

(Invitrogen) at a 2:1 ratio according to the manufacturer's instructions, in antibiotic-free DMEM. After 24 h, cells were transfected again with 1 µg polyinosinic–polycytidylic acid (Poly(I:C), Sigma Aldrich) using Lipofectamine 2000 at a 2:1 ratio in Opti-MEM (Gibco). After a further 24 h, cells were harvested in passive lysis buffer for western blot analysis (Promega) and RNA was extracted by mixing cell lysate 1:3 with TRIzol LS (Invitrogen) then using a Direct-zol RNA MiniPrep kit (Zymo Research).

### Phylogenetics and identification of novel TRS-B sites

For Figs 2B and S2A phylogenetic trees produced by Nextstrain for SARS-CoV-2 global data were downloaded as SVG files under the CC BY license on 24/05/2024 (Fig 2B) or 23/09/2021 (S2A Fig), following annotation to indicate amino acids at position 204 of N, using the "Colour by… genotype" command. To calculate emergence of novel TRS-B sites, a complete snapshot of the GISAID EpiCov database was downloaded on 12 Nov 2021 and the subset of 'complete', 'high coverage' SARS-CoV-2 genomes extracted according to the associated metadata. Using locateTRS (https://doi.org/10.25418/crick.27959910) these were aligned in parallel in batches of 10,000 sequences to the Wuhan-Hu-1 (NC_045512) reference with MAFFT [100] ('mafft –nuc –nwildcard – 6merpair --keeplength –addfragments') and the results merged to produce a complete alignment. Locations of the AAACGAAC and ACGAAC sequence motifs were extracted from individual sequences within the alignment using 'seqkit locate' 42 using regular expressions allowing for alignment gaps ('-' characters) within each search sequence. These were subsequently compared to the locations of the same search sequences within the Wuhan-Hu-1 reference using 'bedtools subtract' to identify novel occurrences and their frequencies calculated with 'bedtools genomecov'. As the '--keeplength' flag to MAFFT carries the potential to create artefactual alignments by removing nucleotides inserted relative to the reference, for each novel location of either sequence motif, the corresponding genomes were extracted and re-aligned to the Wuhan-Hu-1 reference using MAFFT without this flag to ensure that reported sites were valid.

To screen for possible convergent evolution, we examined the Audacity tree made available by the GISAID Initiative [30], using a mutation-annotated tree inferred by UShER [101]. We examined the final phylogenetic tree using Taxonium (https://github.com/theosanderson/taxonium), searching for nodes annotated with the mutations under consideration, and which had more than 30, 50, or 100 descendants. Where clades were found, we considered the alternative possibilities of convergent evolution, phylogenetic misplacement, or artefacts due to contamination with other B.1.1 sequences. In particular we looked for the presence of mutations back to reference, often indicative of artefacts due to contamination or failure to trim primer sequences, and used CoV-Spectrum [29] to examine sub-clade dynamics and prevalence. CoV-Spectrum was also used to determine the proportions of sequences carrying certain mutations in S2D and S2E Fig and Venn diagram representations were generated with BioVenn [102].

### Supporting information

**S1 Fig. Nanopore sequencing of sgmRNA amplicons from human swab samples. (A)** Schematic for detection of N, N.ORF3 and ORF9b-specfic sgmRNA using the endpoint reverse transcription PCR (RT-PCR) assay depicted in Fig 2E–G. **(B)** Nanopore sequencing of endpoint PCR products from N.iORF3-, N-, or ORF9b-specific sgmRNAs in clinical swab samples from EU1 and Alpha lineages, expressed as either raw Nanopore read numbers, determined by TRS-B junction-specific sequences, (left panel) or expressed as a proportion of N reads per sample (right panel). Data are means and standard errors of 12 swab samples per

lineage and *p*-values represent pairwise *t*-tests for each sgmRNA species. TRS, transcription regulatory sequence. Data underlying this figure can be found in: https://doi.org/10.25418/crick.27952842.
(PDF)

**S2 Fig. N.iORF3 and extended homology to TRS-flanking regions have both evolved convergently. (A)** Phylogenetic reconstruction of the Iota variant evolution (B.1.526), highlighting the emergence of N.iORF3 (red). **(B)** A silent mutation at S202 generates extended homology of the N.iORF3 TRS-B region to the 5′UTR region flanking the TRS-L. **(C)** Phylogenetic reconstruction of SARS-CoV-2 evolution in humans, with independent emergences of extended N.iORF3 TRS sequence highlighted (see S1 Table). The phylogenetic tree in panel a was adapted from Nextstrain based on the Iota-focussed build, and in panel C on the Omicron.21K focussed-build [103,104]. **(D)** Proportion of SARS-CoV-2 sequences with the extended N.iORF3 TRS-B sequence by lineage, showing B.1.1 and P.1 (Gamma) in the left panel, and Alpha and four Omicron sub-lineages in the right panel. **(E)** Venn diagram showing proportion of B.1.1 sequences (red) with the A28877U, G28878C mutations (blue), and minimal emergence of these mutations outside of the B.1.1 lineage. TRS, transcription regulatory sequence. Data underlying this figure can be found in: https://doi.org/10.25418/crick.27952842.
(PDF)

**S3 Fig. Convergent evolution of a TRS-B site within the coding region of Spike protein, overlapping the connector domain of Spike/S2. (A)** Schematic of the SARS-CoV-2 genome (upper panel) and frequency of emergence of the TRS-B sequence (AAACGAAC) in the global SARS-CoV-2 population (lower panel). **(B)** Diagram of the Spike ORF, including a potential transframe product. **(C)** Sequence alignment of amino acids 1071−1086 of Spike, and alignment of the corresponding nucleotide sequences show emergence of a new TRS-B sequence. **(D)** The sequence context of the novel Spike.iORF sgmRNA, showing TRS-B (blue highlight), extended homology to the 5′UTR (green highlight) during nascent (−) strand RNA synthesis (black), and downstream tandem start codons and Kozak contexts (yellow highlight). **(E)** Phylogenetic reconstruction of SARS-CoV-2 evolution in humans, with independent emergences of Spike.iORF TRS sequence with ≥50 descendant genomes highlighted in pink (see S1 Table). The schematic and abbreviations of Spike protein domains shown in panel B was adapted from Lan and colleagues [16]. TRS, transcription regulatory sequence.
(PDF)

**S4 Fig. Evolution of minimal TRS-B (minTRS-B) sites upstream of the canonical Envelope minTRS-B. (A)** Schematic of the SARS-CoV-2 genome (upper panel) and frequency of emergence of the minTRS-B sequence (ACGAAC, middle panel) and full-length TRS-B sequence (AAACGAAC, lower panel) in the global SARS-CoV-2 population. **(B)** Diagram of the ORF3a, including a potential transframe product, showing two loci of new minTRS-B emergence (i) and (ii), upstream of (iii) the canonical E minTRS-B. **(C)** Sequence alignment of amino acids 238−275 of ORF3a, and alignment of the corresponding nucleotide sequences show two loci of new minTRS-B emergence. **(D)** The sequence context of the novel (i) ORF3a.iORF1 and (ii) ORF3a.iORF2 sgmRNA relative to (iii) existing minTRS-B that drives canonical E sgmRNA expression, showing minTRS-B sites (blue highlights), extended homologies to the 5′UTR (green highlights) during nascent (−) strand RNA synthesis (black), and downstream start codons and Kozak contexts (yellow highlights). **(E)** Phylogenetic reconstruction of SARS-CoV-2 evolution in humans, with independent emergences of ORF3a.iORF minTRS-B sites with ≥50 descendant genomes highlighted in orange, with overlaid (i)

or (ii) annotation in white denoting ORF3a.iORF1 minTRS-B and ORF3a.iORF2 minTRS-B emergence, respectively. **(F)** Percentage of sequenced Delta variant genomes in Australia that contain (ii) ORF3a.iORF2 minTRS-B mutation (See S1 Table). The plot in panel F was generated using CoV-Spectrum [44]. TRS, transcription regulatory sequence.
(PDF)

**S5 Fig. Validation of sgmRNA reverse transcription qPCR (RT-qPCR) and determination of sgmRNA copy number. (A)** Schematic representation of qRT-PCR primer probe sets for N, N.iORF3 and E sgmRNAs and **(B)** standard curves using synthetic cDNA oligonucleotide templates. Insets show linear regression of Ct plotted against $\log_{10}$-transformed cDNA copy number. **(C)** RT-qPCR of VeroE6 cells infected with B, B.1.1, Alpha or mock cells, or water controls, validating RT-qPCR specificity. Bars represent the mean and standard deviation of three biological replicates. RT-qPCR analysis of N.iORF3 sgmRNA copy number, expressed as a ratio of N copy number in **(D)** clinical swabs, and **(E)** infected VeroE6 cells in culture at 7 and 24 h post-infection. For infected VeroE6 cells, data are means and standard deviations of at least three biological replicates. For clinical swabs, data are means and standard deviations of 4 (EU1/Alpha) or 12 (Delta/Omicron) swab samples per lineage. Data underlying this figure can be found in: https://doi.org/10.25418/crick.27952842.
(PDF)

**S6 Fig. Extended reverse transcription qPCR (RT-qPCR) analysis of sgmRNA expression in SARS-CoV-2 infected clinical swabs and cell culture. (A, B)** N.iORF3 copy number expressed as a percentage of Nucleocapsid copy number. **(C–H)** Expression of Envelope (B), Nucleocapsid (C) or N.iORF3 (D) sgmRNA, normalised to genomic RNA (ORF1ab) and expressed as fold change compared to the control virus condition: for swab samples (A, C, E, G), EU1, and for infections in VeroE6 cells (B, D, F, H), B lineage. Alpha-ins represents an additional Alpoha isolate with an insertion upstream of the Nucleocapsid codiing region, described in S8 Fig. Data underlying this figure can be found in: https://doi.org/10.25418/crick.27952842.
(PDF)

**S7 Fig. ORF9b-specific sgmRNA reverse transcription qPCR (RT-qPCR). (A)** Schematic representation of qRT-PCR primer probe sets for N and ORF9b sgmRNAs and **(B)** standard curve using synthetic cDNA oligonucleotide templates for ORF9b sgmRNA. Standard curve for N sgmRNA is shown in Extended Data Fig 1B. **(C)** Amplification of N (black) or ORF9b (purple) sgmRNA in Vero E6 cells infected with B, B.1.1 or Alpha, in technical duplicate, representative of three biological replicates. **(D)** Gel electrophoresis of RT-PCR products from ORF9b RT-qPCR reactions. Data underlying this figure can be found in: https://doi.org/10.25418/crick.27952842 and https://doi.org/10.25418/crick.27953013.
(PDF)

**S8 Fig. Expression of N.iORF3 protein in infection. (A, B)** Western blot analysis of Mock, B-lineage or Alpha-infected VeroE6 cells, at 24 h post-infection without treatment (A) or at 16 h post-infection with or without 8 h MG-132 treatment (B). **(C)** Western blot analysis of Mock VeroE6 cell or cells infected with reverse-genetics-derived viruses as indicted (see Figs 5 and S9). MW, molecular weight marker. Data underlying this figure can be found in: https://doi.org/10.25418/crick.27953013.
(PDF)

**S9 Fig. sgmRNA expression in reverse-genetics-derived viruses. (A)** Summary of reverse genetics mutants used in the Alpha backbone, showing nucleotide mutations and

corresponding amino acid changes (left panel) and schematic of the experimental design (right panel). **(B–D)** reverse transcription qPCR (RT-qPCR) analysis of the indicated sgmRNAs, normalised to ORF1ab and expressed as fold change relative to expression in Alpha-WT. $Log_{10}$-transfrmed values were compared by one-way ANOVA with Tukey's multiple comparisons test. Data underlying this figure can be found in: https://doi.org/10.25418/crick.27952842.
(PDF)

**S10 Fig.  Growth of WT- and Alpha-backbone virus mutants. (A)** Summary of reverse genetics mutants used in the WT (Wuhan-Hu-1 S:D614G) backbone, showing nucleotide mutations and corresponding amino acid changes (left panel) and schematic of the experimental design (right panel). **(B)** Growth of mutant viruses individually, measured by reverse transcription qPCR (RT-qPCR) against ORF1ab, normalised to actin and **(C)** corresponding area under the curve (AUC) values. Data are means and standard deviations of three biological replicates, compared to WT-N:KR by one-way ANOVA (C). **(D, E)** Head-to-head competition assay comparing fitness Alpha-WT and Alpha-N:RG viruses (D), or Alpha-N:RG and Alpha-silTRS viruses (E), measured by Illumina sequencing of amplicons spanning the N.iORF3 TRS-B region and expressed as percentage of WT-N:KR reads. Total ORF1ab expression, normalised to actin, is shown on the right $y$-axes for reference. Data underlying this figure can be found in: https://doi.org/10.25418/crick.27952842.
(PDF)

**S11 Fig.  Growth of WT- and Alpha-backbone virus mutants in MDA5 KO and RIG-I KO cells. (A–C)** Growth of Alpha-WT (A), Alpha-N:RG (B) or Alpha-silTRS (C) in A549-dual ACE2-TMPRSS2 cells (WT), MDA5 knockout (MDA5 KO) or RIG-I KO cells, measured by reverse transcription qPCR (RT-qPCR) against ORF1ab, normalised to 18S rRNA and **(D)** corresponding area under the curve (AUC) values. Data are means and standard deviations of three biological replicates, compared by two-way ANOVA with Tukey's multiple comparisons. *P*-values are shown. One-way ANOVA comparisons of individual time points for each virus are presented in S3 Table. NB: Growth curve data and AUC values for Alpha-WT in WT cells, Alpha-WT in RIG-I KO cells, Alpha-silTRS in WT cells and Alpha-silTRS in KO cells are the same as presented in Fig 6B and C.
(PDF)

**S1 Table.  Convergent evolution of TRS-B and minTRS-B sites within SARS-CoV-2.** Identified nodes on a phylogenetic reconstruction of SARS-CoV-2 evolution in humans representing clusters of novel TRS and sgmRNA emergence (see Materials and methods). For each emergence event, the parental lineage, approximate date ("Approx. Date") and location ("Location Focus") of circulation are annotated, as well as number of identified descendant genomes within each cluster ("N="), whether it appears to be a true independent emergence event ("Indep?", see Materials and methods) and whether it has further extended homology to the 5′UTR ("+Homol?"). Each cluster is further annotated with potential interrelationships ("Links") and additional notes, including context to its spread based on analysis using CoV-Spectrum [29].
(PDF)

**S2 Table.  One-way ANOVA analysis with Tukey's multiple comparisons test of individual time points for data shown in Fig 5B.** Data were log-transformed, six biological replicates across two independent experiments. *P*-values less than 0.05 are highlighted in bold. hpi, hours post-infection.
(DOCX)

**S3 Table. One-way ANOVA analysis with Tukey's multiple comparisons test of individual time points for data shown in Figs 6B and S10.** Comparisons were carried out for each virus, comparing genome expression at each time point in KO cell lines compared to WT cells. Data were log-transformed, three biological replicates. *P*-values less than 0.05 are highlighted in bold. hpi, hours post-infection.
(DOCX)

**S4 Table. Primer sequences.**
(DOCX)

**S5 Table. GISAID Accession Numbers and Acknowledgements.**
(XLSX)

## Acknowledgments

We thank Simon Caidan, Robert Goldstone, Maria Greco, and Michael Bennett for technical assistance; and Steve Gamblin, George Kassiotis, Wendy Barclay, and Ervin Fodor for helpful discussions. We thank Wendy Barclay, 'Assessment of Transmission and Contagiousness of COVID-19 in Contacts' (ATACCC), the NIHR Health Protection Research Unit in Respiratory Infections, Imperial College London (NIHR200927), Public Health England, Leo James, Steve Goodbourn, Tulio de Oliveira, Alex Sigal, Khadija Kahn, Thushan de Silva, Gavin Screaton, and the G2P-UK National Virology Consortium for support and rapid sharing of reagents and samples, as well as the participants of the Crick SARS-CoV-2 Longitudinal Study: Understanding Susceptibility, Transmission and Disease Severity (Legacy Study). We thank all researchers who have submitted SARS-CoV-2 genomes to the GISAID database: an acknowledgement table for the specific set analysed here can be found in S5 Table. The phylogenetic reconstructions depicted in Figs 2B and S2A are derivatives of those produced by nextstrain.org, used under CC BY. For the purpose of Open Access, the author has applied a CC BY public copyright licence to any Author Accepted Manuscript version arising from this submission.

## Author contributions

**Conceptualisation:** Harriet V. Mears, David L. V. Bauer.

**Data curation:** Theo Sanderson.

**Funding acquisition:** Arvind H. Patel, Massimo Palmarini, Bryan Williams, Sonia Gandhi, Charles Swanton, David L. V. Bauer.

**Investigation:** Harriet V. Mears, George R. Young, Theo Sanderson, Ruth Harvey, Jamie Barrett-Rodger, Rebecca Penn, Vanessa Cowton, Wilhelm Furnon, Giuditta De Lorenzo, Margaret Crawford, Daniel M. Snell, Ashley S. Fowler, Anob M. Chakrabarti, Saira Hussain, Ciarán Gilbride, Edward Emmott, Katja Finsterbusch, Thomas P. Peacock.

**Methodology:** Harriet V. Mears, George R. Young, Theo Sanderson, Anob M. Chakrabarti, David L. V. Bauer.

**Project administration:** Emma Wall, Charles Swanton, David L. V. Bauer.

**Resources:** Theo Sanderson, Katja Finsterbusch, Jakub Luptak, Emma Wall, Sonia Gandhi, Charles Swanton.

**Software:** George R. Young, Theo Sanderson.

**Supervision:** Jérôme Nicod, Bryan Williams, Sonia Gandhi, Charles Swanton, David L. V. Bauer.

**Visualisation:** George R. Young, Theo Sanderson.

**Writing – original draft:** Harriet V. Mears, David L. V. Bauer.

**Writing – review & editing:** Harriet V. Mears, David L. V. Bauer.

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
