## [Editor Report · Decision Letter 0]

29 Jul 2022

Dear Dr. Bauer, 

Thank you for submitting your manuscript entitled "Emergence of new subgenomic mRNAs in SARS-CoV-2" for consideration as a Research Article by PLOS Biology.

Your manuscript has now been evaluated by the PLOS Biology editorial staff and I am writing to let you know that we would like to send your submission out for external peer review.

Once your full submission is complete, your paper will undergo a series of checks in preparation for peer review. After your manuscript has passed the checks it will be sent out for review. To provide the metadata for your submission, please Login to Editorial Manager (https://www.editorialmanager.com/pbiology ) within two working days, i.e. by Jul 31 2022 11:59PM.

Kind regards,

Paula

Senior Editor

PLOS Biology

---

## [Decision Letter · Decision Letter 1]

13 Sep 2022

Dear Dr. Bauer,

Thank you for your patience while your manuscript "Emergence of new subgenomic mRNAs in SARS-CoV-2" was peer-reviewed at PLOS Biology. Your manuscript has been evaluated by the PLOS Biology editors, an Academic Editor with relevant expertise, and by several independent reviewers.

As you will see in the reviewer reports, which can be found at the end of this email along with the comments from the Academic Editor, although the reviewers find the work potentially interesting, they have also raised a substantial number of important concerns. Based on their specific comments and following discussion with the Academic Editor, it is clear that a substantial amount of work would be required to meet the criteria for publication in PLOS Biology. However, given our and the reviewer interest in your study, we would be open to inviting a comprehensive revision of the study that thoroughly addresses all the reviewers' comments and the Academic Editor's comments. Given the extent of revision that would be needed, we cannot make a decision about publication until we have seen the revised manuscript and your response to the reviewers' comments. Your revised manuscript would need to be seen by the reviewers again, but please note that we would not engage them unless their main concerns have been addressed. 

We appreciate that these requests represent a great deal of extra work, and we are willing to relax our standard revision time to allow you 6 months to revise your study. Please email us (plosbiology@plos.org) if you have any questions or concerns, or envision needing a (short) extension.

**IMPORTANT - SUBMITTING YOUR REVISION**

*Resubmission Checklist*

*Published Peer Review*

*PLOS Data Policy*

Please note that as a condition of publication PLOS' data policy (http://journals.plos.org/plosbiology/s/data-availability ) requires that you make available all data used to draw the conclusions arrived at in your manuscript. If you have not already done so, you must include any data used in your manuscript either in appropriate repositories, within the body of the manuscript, or as supporting information (N.B. this includes any numerical values that were used to generate graphs, histograms etc.). For an example see here: http://www.plosbiology.org/article/info%3Adoi%2F10.1371%2Fjournal.pbio.1001908#s5

*Blot and Gel Data Policy*

Sincerely,

Paula

---

Senior Editor

PLOS Biology

REVIEWS:

Reviewer #1: Coronavirus replication.

Reviewer #2: Coronavirus replication/pathogenesis.

Reviewer #1: Overall, the manuscript is well written and describes some interesting data consistent with adaptation of SARS-CoV-2 to humans. Work on SARS-CoV-2 will naturally be of interest to a fairly broad audience and data describing how the virus is changing in response to infection of humans is useful for surveillance, vaccine development and also the study of basic virology. However, some data is lacking to fully support the conclusions drawn and discussion of the findings in the context of the wider CoV field, outside SARS-CoV-2, is sparse.

Specific points:

* The coding capacity of CoVs is increasingly being recognized to be significantly larger than the historically annotated genes. Early work using Northern blot to detect viral transcripts often produced numerous bands in addition to the expected sgRNAs. These were frequently written off as non-specific background bands due to a lack of canonical TRSs needed to generate the additional sgRNAs. However, increasingly, additional transcripts are being described for several different CoVs that must use alternate mechanisms for transcription. The data presented in this work does not rule out the possibility that the N.iORF3 sgRNA is expressed from a minimal, non-canonical TRS in parental strains and mutation to a canonical TRS allows for increased expression, and therefore detection, of the sgRNA. By similarity, this would also apply to other sgRNAs described in the work. Although a canonical TRS will likely give maximum expression of an sgRNA from a given location in the genome, functional non-canonical TRSs have been reported. Including, as acknowledged by the authors, a shorter TRS for the SARS-CoV-2 E sgRNA. Although the N.iORF3 sgRNA could not be detected in EU1 infected cells by end-point or qRT-PCR, evidence of this transcript was shown in supp Fig2b using nanopore sequencing of end point PCR products. This was at a lower level than that seen in Alpha infected cells but that should be expected if a non-canonical TRS is being used. This definitely warrants a fuller description in the text and further investigation. Furthermore, western blot data in Fig 3b clearly shows a band corresponding to N.iORF3 in lanes B and Beta, although these should not express the additional sgRNA. I don't think the possibility that the sgRNA is expressed from a non-canonical TRS and this is mutating to produce a canonical TRS weakens the findings of the work. The virus is clearly changing upon continued replication in humans, including upregulating expression of one or more sgRNA(s). In addition, data describing additional mechanisms by which CoVs increase their coding capacity and around the precise requirements for production of sgRNAs is a fascinating and an understudied area that needs more attention.

* How common are AUGs within kozak consensus upstream of a plausible ORF throughout the genome? For this to be maintained within the virus suggests that there may be an existing mechanism for expression.

* It is obvious that as yet not fully understood elements beyond the TRS itself play a role in regulating TRS usage. Reference to earlier work where additional TRSs have been inserted into a CoV genome to allow generation of a new TRS would be useful (such as work in MHV and IBV). More detailed discussion of TRSs in general would also be beneficial. A significant body of work exists for TGEV but work on MHV and IBV is also relevant and important.

* The possibility that sgRNAs may be functioning at the RNA level rather than as mRNAs is not discussed at all.

Reviewer #2: Reviewer Summary:

In this manuscript, Mears et. al. examine novel sub genomic RNAs (sgRNAs) produced during SARS-CoV-2 infection. The authors focus their efforts on the characterization of one novel sgRNA created by the R203K, G204R mutation, which is present in the Alpha, Gamma, and Omicron variants of concern. At the nucleotide level, the R203K, G204R mutation creates a novel Transcription Regulatory Sequence (TRS), resulting in the production of a sgRNA encoding N.iORF3, a putative protein consisting amino of acids 210-419 of SARS-CoV-2 Nucleocapsid. Mears et al. also provide some evidence that other novel sgRNAs may be present in circulating SARS-CoV-2, the significance of which merit further study.

While this reviewer will refrain from commenting on the phylogenetic/bioinformatics portion of this study, there are unfortunately several major issues with the rest of the manuscript as written. First, the authors explicitly brush aside other studies examining both the R203K, G204R and analogous mutations (R203M in delta) at the protein level, thus failing to put their own results into its proper context. From a technical/experimental standpoint, several inadequacies are also present in the author's characterization of N.iORF3. While data is presented confirming a novel sgRNA is being produced, evidence that it is translated into a novel protein is weak. Furthermore, the authors attempt to ascribe a role for N.iORF3 in inhibition of the IFN response, yet do not provide sufficient evidence that this would be true during SARS-CoV-2 infection. Finally, the presence of this novel sgRNA has already been reported by other studies, weakening the novelty of this manuscript given the previously mentioned shortcomings regarding characterization.

Major Issues:

Issue 1:

Line 6 - 7: "While the latter mutations (R203K, G204R) appear unremarkable at the protein level." and lines 32-33: "R203K,G204R mutations appear unremarkable on an amino acid level."

These statements are completely discordant with the current literature, which strongly suggest that the R203K, G204R mutation is absolutely having an effect on the protein level. Specifically, Johnson et al. (PMID: 35728038) provides data suggesting the R203K, G204R mutation alters the phosphorylation of nucleocapsid during infection of Vero and Calu3 cells, Wu et al. (PMID: 34822776) proposes a structural model by which the mutation may alter nucleocapsid's binding to RNA, and Zhao et. al. (PMID: 33837182) demonstrate that R203K, G204R modulates phase separation of the nucleocapsid protein. The authors should discuss why these past studies are wrong, discuss the introduction of a TRS site in the context of these other functions, or else remove such a strong and incongruent statement from the abstract and main text.

Issue 2:

Line 103 -104 and Figure 3b and 3c: "we also observed 103 a band at ~25 kDa in B.1.1 and Alpha-infected cells (Fig. 3b), consistent with the mobility of 104 recombinant N.iORF3 protein (Fig. 3c)"

If the authors wish to suggest In Figure 3b that the lower molecular weight bands are consistent in size to recombinant N.ORF3, then recombinant N.ORF3 should be ran on the same gel. Referring to a different figure (Figure 3c) is not sufficient. Additionally, at least one study suggests that nucleocapsid undergoes extensive proteolytic cleavage (PMID: 34548480). The authors need to discuss this as an alternative explanation for their results, especially given the numerous other bands of equal or greater intensity throughout the gel. Finally, by this reviewer's eye a faint N.iORF3 band seems to be present in the Beta variant, despite it lacking the R203K, G204R mutation. This band's presence in the Beta variant is in direct conflict with the authors' own conclusion: that this band represents the N.iORF3 protein is being translated from a novel sgRNA produced by the R203K, G204R mutation. More evidence that N.iORF3 is actually being expressed a detectable levels is needed.

Issue 3:

Lines 104 - 113, and Figure 3c and 3d: The authors own data suggests that N.iORF3 is expressed at low levels relative to full length N (RNA - Extended Data 1d-e, Protein - Figure 3b). Additionally, SARS-CoV-2 infection is known to be more sensitive to IFN compared to SARS-CoV-1 (PMID: 32938761). With this in mind, merely showing a modest effect on ISG induction when overexpressing N.iORF3 and poly-ic in HEK293 cells is unconvincing given its artificiality. At minimum, the authors need to perform additional experiments showing that the R203K, G204R mutation provides resistance to IFN relative to wildtype during SARS-CoV-2 infection. Ideally, the authors would also link this resistance to the presence of N.iORF3.

Issue 4:

Discussion: Related to issue 1 above, several analogous mutations at this locus SARS-CoV-2 nucleocapsid, including the R203M mutation in delta and T205I mutation in beta (see authors own Figure 1C). These mutations have dominated individual lineages (and the total circulating SARS-CoV-2 in general in the case of R203M/Delta) at various points in the pandemic, yet they do not create novel sub genomic transcripts. While this does not preclude a novel sgRNA and N.iORF3 contributing to the (previously reported) enhanced fitness of the R203K, G204R mutation, it implies that it may not be necessary and strongly suggested is isn't the sole contributor. The authors need to address why these other mutations emerged and are alone sufficient to enhance replication/fitness without producing novel sgRNA (PMID: 34735219). 

Issue 5:

Novelty: Other studies have already been published suggesting a new sgRNA is produced by the R203K, G204R mutation (PMID: 34541432). Given the other shortcomings, it is hard to see how merely confirming the presence of this transcript with a different methodology (RT-PCR/public data sets vs deep sequencing of clinical samples) provides any new information. The authors should focus on strengthening the other aspects of their manuscript to increase its novelty.

Minor Issues:

Issue 6:

Line 72/ Figure 2b: The clarity of these gels would increase significantly if the ladder was labeled base with pair markers.

Issue 7:

Line 87 - 97: While the authors RT-qPCR assays do indicate a N.iORF3 sub-genomic assay is present at the RNA level with ~1% the expression of full length N, this review wonders if a Northern Blot would not be more convincing.

COMMENTS FROM THE ACADEMIC EDITOR:

My biggest critique is that the current organization is a bit awkward. They begin with a very narrow focus on N.iORF3, then they present a broader search for other similar mutations and wrap up by describing some of the other variants that are common. It seems to me that the flow would be improved if they began with their broad search for new TRS-B sites (ED-Fig 4a), then continued with further characterization of the three most common variants they observed. I think the overall content would mostly be unchanged, but the flow would be more natural.

Another aspect that I found a bit confusing was why they primarily refer to these substitutions by their AA-level designations when it is really the nucleotide-level substitutions that are important for their findings. For example, they talk about "R203K, G204R" as a double substitution. However, there are really three nt-level substitutions that underly these AA-level changes. I understand that the community has been much more focused on the AA-level changes, but I would encourage the authors to embrace the nt-level designations as the primary level of description in their paper.

Additional comments:

1. Fig. 1a: the legend says that the lineage defining mutation for B.1 is indicated, but it's not.

2. Fig. 1a: I believe the "K204R" reference in the legend is a typo.

3. The authors need to explain why N.iORF3 expression was compared against E expression (Fig. 2d).

4. Fig. 2b legend should include a description for all of the labels.

5. Manuscript needs a better description of why Flu-A NS1 is an appropriate positive control.

6. The manuscript needs more context for the analysis shown in Fig. 3d. Yes, N.iORF3 can act as an innate immune signaling antagonist, BUT it's activity seems to be lower than that of full-length N and it's expressed at a much lower level. So, would this activity really be expected to play a substantial role during infection?

7. Lines 20 and 22, refs should be provided for the statements about the frequencies of these different mutations.

8. Lines 84-85: rephrase "but not from the B.1 lineage." Isn't B.1.1 also part of the B.1 lineage?

---

## [Decision Letter · Decision Letter 2]

14 Aug 2024

Dear Dr Bauer,

Thank you for your patience while we considered your revised manuscript "Emergence of new subgenomic mRNAs in SARS-CoV-2" for publication as a Research Article at PLOS Biology. Your revised study has been evaluated by the PLOS Biology editors, the Academic Editor and the original reviewers.

In light of the reviews, which you will find at the end of this email, we would like to invite you to revise the work to thoroughly address the reviewers' reports.

As you will see below, the reviewers acknowledge the significant improvements made to the study, but some concerns have emerged during this second revision. Reviewer #1 requests confirmation of certain observations, such as the effect of mutations on sgRNA expression, and additional controls. Reviewer #2 remains unconvinced about the evidence regarding the function of the truncated protein and its role in IFN signaling. While the reviewers have noted that the current work would be suitable for publication in a subject-relevant journal, a broader-interest journal would require a deeper functional understanding of the resulting protein.

IMPORTANT: After discussions with the Academic Editor and reviewers, it is necessary to conduct the experiment suggested by Reviewer #2 to strengthen the evidence regarding the truncated protein's function. If the results indicate potential redundancies, please present these findings and discuss possible alternative functions of the protein. This is required for acceptance. While we believe addressing point 3 of Reviewer #1 would enhance the study, it is not mandatory for publication.

Given the extent of revision needed, we cannot make a decision about publication until we have seen the revised manuscript and your response to the reviewers' comments. Your revised manuscript is likely to be sent for further evaluation by all or a subset of the reviewers.

**IMPORTANT - SUBMITTING YOUR REVISION**

*Re-submission Checklist*

*Published Peer Review*

*PLOS Data Policy*

Please note that as a condition of publication PLOS' data policy (http://journals.plos.org/plosbiology/s/data-availability ) requires that you make available all data used to draw the conclusions arrived at in your manuscript. If you have not already done so, you must include any data used in your manuscript either in appropriate repositories, within the body of the manuscript, or as supporting information (N.B. this includes any numerical values that were used to generate graphs, histograms etc.). For an example see here: http://www.plosbiology.org/article/info%3Adoi%2F10.1371%2Fjournal.pbio.1001908#s5

*Blot and Gel Data Policy*

Sincerely,

Melissa

Melissa Vazquez Hernandez, Ph.D.

Associate Editor

PLOS Biology

REVIEWERS' COMMENTS: 

Reviewer #1: 

The updated manuscript addresses the queries raised in the last review providing a more comprehensive analysis of control of expression of the novel sgmRNA and a more detailed discussion in the context of previous knowledge. There are still some outstanding comments/questions and the discussion could still be expanded a little:

1. Line 62-63: Although the spectrum of sgRNAs identified by this approach may overrepresent what is truly made, it is misleading to gloss over the fact that some viruses make what have only recently been recognised as functional sgmRNAs due to their use of a minimal TRS (and actually quite efficiently). See data on IBV.

2. Fig 5. Did you confirm that these mutations abolish sgRNA expression?

3. Alpha-silTRS - It would be most informative to include a control where the minimal TRS (GCACA) is mutated on its own. And also, there is no confirmation that it is not the different codon usage for the KR that's the problem here.

4. Fig 5H - why not test this with the RG viruses that you've made to link back to biological relevance? I appreciate that redundancy in viral antagonists of IFN induction/signalling may mask the effect but it's still a valuable experiment to include.

5. 316-7 - I strongly query the legitimacy of calling this a "novel" sgmRNA specific assay. This has been done for other CoVs in the past.

6. Lines 396-397 - I think it worth expanding the discussion here. It has also been found that a minimal TRS is actually more efficient/preferentially used over a new inserted complete TRS. There may be differences in how much flexibility is tolerated between the CoV genera.

7. Lines 407-409 - Isn't this surprising though? Would you propose then that the overall transcriptional output increases to keep the other sgRNA levels the same AND include expression of this new sgRNA? Maybe comparing additional sgRNA levels would be useful here?

Reviewer #2: 

This manuscript form Mears et al. details the how mutations among SARS-CoV-2 variants lead to expression of novel sub-genomic transcripts via the creation of new transcription regulatory sequences (TRSs). While evidence from publicly available datasets is examined demonstrating the frequency at which these novel TRSs are created, the authors focus their efforts on the characterization of two: one within Nucleocapsid created through the R203K, G204R substitution (creating a novel N.iORF3) and a second within ORF1ab (creating a novel nsp16.iORF1). Data is presented demonstrating the expression of these novel transcripts at the mRNA and protein levels among clinical isolates, viral stocks, and viruses created using reverse genetics.

Compared to the authors initial submission, this version of the manuscript is greatly improved. Summarizing the authors' specific claims:

Major Points with strong supporting evidence:

* Creation of novel TRS sites are not isolated events (Fig. 2A)

* The novel N.iORF3 is expressed at the RNA (Fig. 2F, Fig. 4A-D)) and protein (Fig. 4E) levels

* The protein band corresponding with N.iORF3 is specifically caused by the R203K, G204R mutation (Fig. 5B)

* N.iORF3 expression enhances viral replication and fitness, and its removal of the TRS by reverse genetics attenuates the Alpha variant even while maintaining the mutation on the amino acid level (Fig. 5C-E, Fig. S9),

* Nsp16.iORF1 is expressed at the RNA level (Fig. 3E).

Overall, these 5 points are very well supported both by the figures cited above as well as the supplemental material. However, the authors attempt to make a sixth claim regarding the function of N.iORF3 in inhibiting IFN signaling, which I still believe lacks sufficient evidence. This point is elaborated on below.

Major Weaknesses

* Presented data inadequately support the claim the N.iORF3 functions to inhibit Type I IFN signaling.

While N.iORF3 is convincingly playing some role in viral replication and fitness (Fig. 5C-E, Fig. S9), I remain unconvinced that N.iORF3 functions to inhibit IFN signaling during infection. To support this claim, the authors ectopically express SARS2 N and SARS2 N.iORF3 in 293T cells, transfect poly:iC, and measure the induction of IFNB and IFTIT mRNA (Fig. 5F-H). I have several lingering critiques of this experiment, all of which I stated in my initial review. First, according to the figure legend and the western blot data in Fig. 5H, SARS2 N and SARS2 N.iORF3 are being expressed at equal amounts. However, at each concentration N.iORF3 appears to less effectively inhibit IFNB expression relative to full length SARS2 N (though the authors do not directly compare the two with statistics). However, the authors own data suggests that N.iORF3 expression is approximately 100-fold lower than full length SARS 2N (Fig. 4D) during infection. Taken together, these data may me skeptical that N.iORF3's primary role is the inhibition of Type I IFN. To summarize succinctly, while the data in Fig. 5H suggests that ectopic expression of N.iORF3 can inhibit IFN, it may have less of an effect compared full length SARS2 N, and due to lower levels of expression it seems unlikely that it would have this effect during infection. 

A relatively straightforward experiment can be performed to address this issue. If N.iORF3's major role is the inhibition of interferon signaling, then a virus expressing N.iORF3 should be more resistant to interferon treatment relative to one that does not. Fortunately, the authors already have all the tools needed to test this. I would suggest that the Alpha-WT, Alpha N:RG, and Alph-siTRS be used to infect Vero-E6 TMPRSS2 cells with and without IFN pretreatment. If the authors' current hypothesis is correct, that Alpha-N:RG and Alpha-siTRS should be more sensitive to Interferon treatment when compared with Alpha-WT. For an example of this sort of assay, see Fig. 1 of Lokugamage et al. (PMID: 32938761). If not, then it is likely that N.iORF3 has a different function. This is a fairly easy experiment to perform and would greatly strengthen the authors assertion that N.iORF3 functions to inhibit interferon signaling during infection.

---

## [Decision Letter · Decision Letter 3]

20 Nov 2024

Dear Dr Bauer,

Thank you for your patience while we considered your revised manuscript "Emergence of new subgenomic mRNAs in SARS-CoV-2" for publication as a Research Article at PLOS Biology. This revised version of your manuscript has been evaluated by the PLOS Biology editors, the Academic Editor and one of the original reviewers.

Based on the reviews and on our Academic Editor's assessment of your revision, we are likely to accept this manuscript for publication, provided you satisfactorily address the remaining editorial points. Please also make sure to address the following data and other policy-related requests.

a) We routinely suggest changes to titles to ensure maximum accessibility for a broad, non-specialist readership, and to ensure they reflect the contents of the paper. In this case, we would suggest a minor edit to the title, as follows. Please ensure you change both the manuscript file and the online submission system, as they need to match for final acceptance:

"Emergence of SARS-CoV-2 subgenomic mRNAs that that enhance viral fitness and immune evasion"

b) The Ethics statement needs to be a separate, independent (and the first) subheading in the Material & Methods section. We also require a statement that the participants provided consent either in written or oral form.

https://journals.plos.org/plosbiology/s/ethical-publishing-practice

c) You may be aware of the PLOS Data Policy, which requires that all data be made available without restriction: 

http://journals.plos.org/plosbiology/s/data-availability . For more information, please also see this editorial: http://dx.doi.org/10.1371/journal.pbio.1001797

Please supply the numerical values either in the a supplementary file or as a permanent DOI’d deposition for the following figures:

Figure 4BCD, 5CDE, 6BCF, S1B, S2DE, S5BC, S6A-H, S7BC, S9BCD, S10BCDE, S11ABCD. Please note that Figure S3 is missing

d) Please cite the location of the data clearly in all relevant main and supplementary Figure legends, e.g. “The data underlying this Figure can be found in S1 Data” or “The data underlying this Figure can be found in https://doi.org/10.5281/zenodo.XXXXX”

e) We require the original, uncropped and minimally adjusted images supporting all blot and gel results reported in the Figures 2F, 3E, 4E, 5B, 6AE, S7D, S8ABC

We will require these files before a manuscript can be accepted so please prepare and upload them now. Please carefully read our guidelines for how to prepare and upload this data: https://journals.plos.org/plosbiology/s/figures#loc-blot-and-gel-reporting-requirements

f) Please also provide the tree files for Figures 2B, 3F, S2AC, S4E

g) Please ensure that your Data Statement in the submission system accurately describes where your data can be found and is in final format, as it will be published as written there.

h) Per journal policy, if you have generated any custom code during the course of this investigation, please make it available without restrictions upon publication. Please ensure that the code is sufficiently well documented and reusable, and that your Data Statement in the Editorial Manager submission system accurately describes where your code can be found.

We expect to receive your revised manuscript within two weeks. 

*Published Peer Review History*

*Press*

Sincerely,

Melissa

Melissa Vazquez Hernandez, Ph.D.

Associate Editor

PLOS Biology

REVIEWERS'S COMMENTS:

Reviewer #2: Having read through this manuscript, I feel that the reviewers have addressed all of my concerns. I think that it is now acceptable for publication.

---

## [Editor Report · Decision Letter 4]

11 Dec 2024

Dear Dr Bauer,

Thank you for the submission of your revised Research Article "Emergence of SARS-CoV-2 subgenomic mRNAs that enhance viral fitness and immune evasion" for publication in PLOS Biology. On behalf of my colleagues and the Academic Editor, Jason Ladner, I am pleased to say that we can in principle accept your manuscript for publication, provided you address any remaining formatting and reporting issues. These will be detailed in an email you should receive within 2-3 business days from our colleagues in the journal operations team; no action is required from you until then. Please note that we will not be able to formally accept your manuscript and schedule it for publication until you have completed any requested changes.

PRESS

We also ask that you take this opportunity to read our Embargo Policy regarding the discussion, promotion and media coverage of work that is yet to be published by PLOS. As your manuscript is not yet published, it is bound by the conditions of our Embargo Policy. Please be aware that this policy is in place both to ensure that any press coverage of your article is fully substantiated and to provide a direct link between such coverage and the published work. For full details of our Embargo Policy, please visit http://www.plos.org/about/media-inquiries/embargo-policy/ .

Sincerely, 

Melissa

Melissa Vazquez Hernandez, Ph.D., Ph.D.

Associate Editor

PLOS Biology
